# Towards improved analysis of short mesoscale sea level signals from satellite altimetry

Yves Quilfen[1], Jean-François Piolle[1], Bertrand Chapron[1]

[1]Laboratoire d'Océanographie Physique et Spatiale (LOPS), IFREMER, Univ. Brest, CNRS, IRD, IUEM, Brest, France

*Correspondence to*: Yves Quilfen (yquilfen@ifremer.fr)

**Abstract**. Satellite altimeters routinely supply sea surface height (SSH) measurements, which are key observations for monitoring ocean dynamics. However, below a wavelength of about 70 km, along-track altimeter measurements are often characterized by a dramatic drop in signal-to-noise ratio, making it very challenging to fully exploit the available altimeter observations to precisely analyze small mesoscale variations in SSH. Although various approaches have been proposed and applied to identify and filter noise from measurements, no distinct methodology has emerged for systematic application in operational products. To best address this unresolved issue, the Copernicus Marine Environment Monitoring Service (CMEMS) actually provides simple band-pass filtered data to mitigate noise contamination of along-track SSH signals. More innovative and suitable noise filtering methods are thus left to users seeking to unveil small-scale altimeter signals. As demonstrated here, a fully data-driven approach is developed and applied successfully to provide robust estimates of noise-free Sea Level Anomaly (SLA) signals. The method combines Empirical Mode Decomposition (EMD), to help analyze non-stationary and non-linear processes, and an adaptive noise filtering technique inspired by Discrete Wavelet Transform (DWT) decompositions. It is found to best resolve the distribution of SLA variability in the 30-120 km mesoscale wavelength band. A practical uncertainty variable is attached to the denoised SLA estimates that accounts for errors related to the local signal-to-noise ratio, but also for uncertainties in the denoising process, which assumes that the SLA variability results in part from a stochastic process. For the available period, measurements from the Jason-3, Sentinel-3 and Saral/AltiKa missions are processed and analyzed, and their energy spectral and seasonal distributions characterized in the small mesoscale domain. In anticipation of the upcoming SWOT (Surface Water and Ocean Topography) mission data, the SASSA data set (Satellite Altimeter Short-scale Signals Analysis, Quilfen and Piolle, 2021) of denoised SLA measurements for three reference altimeter missions already yields valuable opportunities to evaluate global small mesoscale kinetic energy distributions.

Keywords: Altimeter Measurement Noise; Empirical Mode Decomposition; Sea Level Mesoscale Variability

## 1 Introduction

Satellite altimetry fosters studies related to ocean dynamics for more than 25 years, often looking to push the limits of these observations to capture ocean motions at ever smaller scales. New paradigms are thus emerging from this observational effort, among them the distinction between balanced and unbalanced motions that can lead to characteristic changes in SSH signal variations and associated spectrum in the 200-30 km wavelength range (e.g. Fu, 1983; Le Traon et al., 2008; Dufau et al., 2016; Tchibilou et al., 2018) or the role of upper-ocean submesoscale dynamics that is critical to the transport of heat between the ocean interior and the atmosphere (Su et al., 2018). One of the main limitations is that altimetry measurements are often characterized by a low signal-to-noise ratio (SNR), which has a significant impact on geophysical analysis capability at spatial scales smaller than 120 km. Main sources of noise are induced by instrumental white noise, errors related to processing, including the re-tracking algorithm and corrections, and errors related to the intrinsic variability of radar echoes in the altimeter footprint that causes the notorious spectral hump in 20Hz and 1Hz data (Sandwell and Smith, 2005; Dibarboure et al., 2014). Furthermore, because retrieved parameters are obtained from the same waveform re-tracking algorithm, they have highly correlated errors, i.e., the standard MLE4 processing produces four estimated parameters with correlated errors (SSH, significant wave height, sigma0, and off-nadir angle). These errors directly limit the accuracy of the SSH measurement, requiring advanced denoising techniques (Quartly et al., 2021).

Analysis of fine-scale ocean dynamics therefore requires preliminary noise filtering, and low-pass or smoothing filters (e.g., Lanczos, running mean, or Loess filter) are frequently used. These filters effectively smoothen altimeter signals, but result in the systematic loss of small-scale ( $< \sim 70$ km) geophysical information, only remove high-frequency noise, and can produce artifacts in the analyzed geophysical variability. Considering that a major source of high-frequency SSH errors is associated to the correlation between SSH and significant wave height (SWH) errors through the retracking algorithm and sea-state bias correction, recent approaches propose to derive a statistical correction to mitigate correlated high-frequency SSH errors (Zaron and de Carvalho, 2016; Quartly, 2019; Tran et al., 2021). Both the spectral hump and white noise variance at 20Hz are indeed significantly reduced. Yet, these approaches are based on the assumption that a separation scale between SWH noise and SWH geophysical information can be defined, to apply a low-pass filter on SWH along-track signals and to estimate the correlated SSH and SWH errors at high-frequency. In practice, a cut-off wavelength of 140 km is used in Tran et al. (2021). However, this approach has a fundamental caveat, as wave-current interactions strongly imprint surface waves and current dynamics at short mesoscale and submesoscale wavelengths (e.g. Kudryavtsev et al., 2017; Ardhuin et al., 2017; McWilliams, 2018; Quilfen et al., 2018; Quilfen and Chapron, 2019; Romero et al., 2020; Villas Bôas et al., 2020). Wave-current interactions are ubiquitous phenomena, and current-induced SWH variability at scales smaller than 100 km can be expected depending upon the strength of the current gradient relative to the wavelength and direction of propagation of surface waves. Applying a correction based on the assumption that SWH variability is primarily all noise below 100 km will therefore likely affect small mesoscale SSH signals in various and complex ways.

To overcome these difficulties, an adaptive noise removal approach for satellite altimeter measurements has been derived. It is based on the non-parametric Empirical Mode Decomposition (EMD) method developed to analyze non-stationary and non-linear signals (Huang et al., 1998; Huang and Wu, 2008). EMD is a scale decomposition

of a discrete signal into a limited number of amplitude and frequency modulated functions (AM/FM), among which the Gaussian noise distribution is predictable (Flandrin et al., 2004). Noise removal strategies can then be developed with results often superior to wavelet-based techniques (Kopsinis and McLaughin, 2009). An EMD-based technique was successfully applied to altimetry data to more precisely analyze along-track altimeter SWH measurements to map wave-current interactions (Quilfen et al., 2018; Quilfen and Chapron, 2019), known to predominate at scales smaller than 100 km. Especially, the method is suitable for processing non-stationary and non-linear signals, and thus for accurate and consistent recovery of strong gradients and extreme values. Building on local noise analysis, the denoising of small mesoscale signals is performed on an adaptive basis to the local SNR. A detailed description of the EMD denoising approach applied to satellite altimetry data is given in Quilfen and Chapron (2021).

In this paper, the method is extended to more thoroughly evaluate an experimental data set of denoised SLA measurements, from three reference altimeters, the Jason-3, Sentinel-3 and Saral/AltiKa, to capture short mesoscale information. Section 2 provides a description of the data sets used and Sect. 3 describes the denoising methodology main principles. In Sect. 4, which presents the results, examples of denoised SLA signals are given, and the energy spectral and seasonal distributions of denoised measurements are characterized in the small mesoscale domain for these three altimeters. Section 5 presents key features for comparison with other distributed datasets that make our approach more attractive. A discussion follows to analyze the main results and a summary is given. Appendices A and B provide details on the denoising scheme and power spectral density calculation, respectively.

## 2 Data

The Copernicus Marine Environment Service (CMEMS) is responsible for the dissemination of various satellite altimeter products, among which the level 3 along-track sea surface heights distributed in delayed mode (product identifier: SEALEVEL_GLO_PHY_L3_REP_OBSERVATIONS_008_062) is the state-of-the-art product that takes into account the various improvements proposed in the frame of the SSALTO/DUACS activities (Taburet et al., 2021). The input data quality control verifies that the system uses the best altimeter data. From these products, which include data from all altimetry missions, we use the "unfiltered SLA" variable to derive our analysis of SLA measurements.

The present study aims to provide research products, the SASSA (Satellite Altimeter Short-scale Signals Analysis) dataset, and innovative solutions for a better exploitation of the mesoscale mapping capabilities of altimeters. The analysis is therefore limited to three current altimeter missions, Jason-3, Sentinel-3, and Saral/AltiKa, each carrying an instrument with particular distinctive characteristics. The Jason-3 altimeter is the reference dual-frequency Ku-C instrument and is used as the reference mission for cross-calibration with other altimeters to provide consistent products in the CMEMS framework. The Satellite for ARgos and ALtika (SARAL) mission carries the AltiKa altimeter, which makes measurements at higher effective resolution due to a smaller footprint obtained in Ka-band (8 km diameter vs 20 km on Jason-3) and a higher pulse repetition rate. The altimeter on board Sentinel-3 is a dual-frequency Ku-C altimeter that differs from conventional pulse-limited altimeter in that it operates in Delay Doppler mode, also known as Synthetic Aperture Radar Mode (SARM). SARM is the primary

mode of operation which provides ~ 300 m resolution along the track. The SAR mode reduces instrumental white noise and is free of "bump" artifact, which is caused by surface backscatter variabilities, blooms and rain-induced inhomogeneities in the LRM footprint, adding spatially coherent error to the white noise (Dibarboure et al., 2014). Still, current SARM measurements are affected by colored noise, which is likely attributed to the effects of swell on SARM observations (Moreau et al., 2018; Rieu et al., 2021). For the CMEMS data set version available at the time of this study, the retracking used to process the data is MLE-4 for Jason-3 and AltiKa and SAMOSA for Sentinel-3 (Taburet et al., 2021). Each altimeter makes measurements at nadir along the satellite track, and the standard CMEMS processing provides data at 1 Hz with a ground sampling that varies slightly from 6 to 7 km depending on the altimeter. At this ground sampling, the average noise affecting the range measurements is different for each altimeter, with SARAL and Sentinel-3 showing significantly lower level of noise than Jason-3 (Taburet et al., 2021). The data set available at CMEMS for the current analysis covers the period until June 2020 with a beginning in March 2013, June 2016, and May 2016 for AltiKa, Sentinel-3, and Jason-3, respectively.

Although only the quality-controlled CMEMS data are used as input in our analysis, ancillary data are useful to support the analysis of SLA data. Indeed, since some of the larger non-Gaussian SLA errors, correlated with high sea state conditions and rain or slick events, are expected to remain after the EMD analysis, SWH and radar cross-section (sigma0) are also provided in the denoised SLA products to allow for further data analysis and editing. These are provided by the sea state Climate Change Initiative (CCI) products, developed by the European Space Agency (ESA) and processed by the Institut Français de Recherche pour l'Exploitation de la Mer (IFREMER, Dodet et al., 2020).

## 3 Methods

The proposed denoising technique essentially builds on the EMD technique (Huang, 1998; Wu and Huang, 2004; Huang and Wu, 2008), and its filter bank characteristics when applied to Gaussian noise (Flandrin et al., 2004). The technique was first adapted to process satellite altimeter SWH measurements (Quilfen et al., 2018; Quilfen and Chapron, 2019; Dodet et al., 2020), and the algorithm is described in detail in Quilfen and Chapron (2021). For the processing of SLA data analyzed in this study, only limited modifications were made, and the algorithm is only briefly described below.

Three main elements characterize the properties of the denoising algorithm: 1) the EMD algorithm that adaptively splits the SLA signal on an orthogonal basis without having to conform to a particular mathematical framework; 2) the denoising algorithm that relies on high-frequency local noise recovery and analysis; 3) an ensemble average approach to estimate a robust denoised SLA signal and its associated uncertainty.

### 3.1 The EMD algorithm

EMD is a data-driven method, often used as an alternative basis to wavelets for denoising a wide variety of signals. EMD decomposes a 1D signal into a set of amplitude- and frequency-modulated components, called intrinsic modulation functions (IMF), which satisfy the conditions of having zero mean and a number of extrema equal to (or different by one) the number of zero crossings. IMFs are obtained through an iterative algorithm, called sifting, which extracts the high-frequency component by iteratively computing the average envelope from the extrema

points of the input signal. The sifting algorithm is first applied to the input SLA signal to derive the first IMF, IMF1, which is removed from the SLA signal to obtain a new signal on which the process is repeated until it converges when the last calculated IMF no longer has a sufficient number of extrema. The original signal is exactly reconstructed by adding all the IMFs. Figure 1 shows two sets of IMF for two passages of SARAL over the Gulf Stream area. The top panels show the two SLA signals and the associated SWH signals for reference (red curves), and the other panels display the full set of IMF (6 and 4 derived IMFs for these two cases, a number that can vary with signal length and observed wavenumber spectrum). Shown in the right panels, IMF1 genuinely maps the high-frequency noise in term of amplitude and phase, which can provide a direct approach to help remove high-frequency noise from  the SLA signal. Local analysis of this high-frequency noise is used to predict and remove the lower frequency noise embedded in other IMFs, as detailed farther. Left panels, IMF1 is also associated with high-frequency noise but shows non-stationary noise statistics that are related to changes in mean sea state conditions. As expected, the high frequency noise of SLA increases with significant wave height. These two examples are general cases, but IMF1 can also contain geophysical information in cases  where the SNR is locally very high, for example in the presence of very large geophysical gradients, or can show the signature of outliers related to the so-called spectral hump (rain, slicks, …). Indeed, depending on the SNR and in specific configurations for the numerical sifting algorithm, the type of large SLA gradient signature shown in IMF2 (right panel) can very well show up in IMF1, in case there is no detectable extrema between measurements number 2 to 8. Since IMF1 analysis is at the heart of the denoising strategy described below, careful preprocessing of IMF1 is necessary before denoising the full signal.

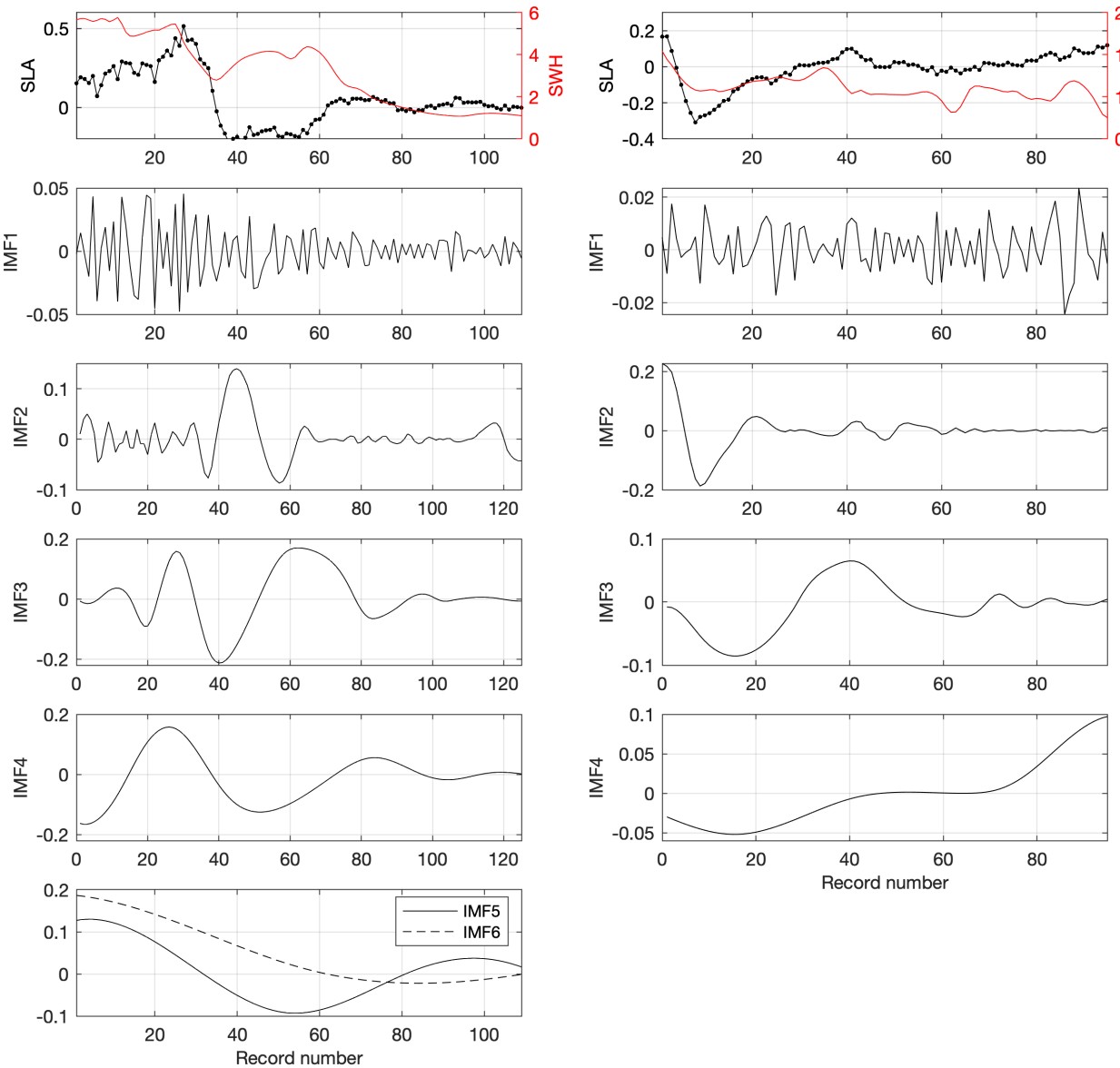

175

**Figure 1: SARAL Sea Level Anomalies (SLA, top panels) in the Gulf Stream area for cycle 106 and passes 53 (left panels) and 597 (right panels), and the corresponding IMFs obtained from the EMD decomposition (second top to bottom panels). The Significant Wave Height (SWH) associated with the SLA is shown in top panels (red curves). All units in meters.**

180

### 3.2 The denoising scheme

Flandrin at al. (2004) applied EMD to a Gaussian noise signal, to demonstrate that the IMF1 has the characteristics of a high-pass filter while the higher order modes behave similarly to a dyadic low-pass filter bank, for which, as

185    they move down the frequency scale, successive frequency bands have half the width of their predecessors. Unlike Fourier or wavelet decompositions for which the noise variance is independent of the scale, the noise contained in each IMF is now "coloured" with a different energy level for each mode. Flandrin et al. (2004) deduced that the variance of the Gaussian noise projected onto the IMF basis can be modeled as follows for the low-pass filter bank:

$$var(h_n(t)) \sim 2^{(\alpha-1)n} \tag{1}$$

where $h_n(t)$ are the IMFs of rank n > 1 and $\alpha$ depends on the Hurst exponent H of the fractional Gaussian noise. For the altimeter dataset, the white noise assumption is made, following studies showing a quasi-white noise spectrum (Zaron and de Carvalho, 2016; Xu and Fu, 2012; Sandwell and Smith, 2005). It corresponds to H=0.5 and $\alpha = 0$ in Eq. (1).

Flandrin et al. (2004) then numerically derive, using Eq. (1) and for different values of H, the relationship between the IMF's variance $E_n$, for n > 1, and the variance of IMF1, $E_1$. For a white noise, this gives:

$$E_n = \frac{E_1}{0.719} 2.01^{-n} \tag{2}$$

With the EMD basis, the noise energy decreases rapidly with increasing IMF rank: ~ 59, 20.5, 10.3, 5.2, 2.6% of the total energy for the top five IMFs, respectively. The first four IMFs account for ~ 95% of the noise energy. Eq. (2) therefore gives the expected noise energy in each IMF to determine the different thresholds below which signal fluctuations can be associated with noise. The threshold formulation introduces the constant factor $A$, which is a control parameter that can be adjusted for different altimeters depending on their characteristic noise levels:

$$T_n = A\sqrt{E_n} \tag{3}$$

with n being the rank of the thresholded IMF.

A detailed description of the entire denoising scheme can be found in Quilfen and Chapron (2021) and the main steps are given in Appendix A.

Figure 2 provides illustrations of the general approach taken to denoising SLA signals. They show the power spectrum density (PSD) of SLA (black curves), the associated IMFs (blue curves), and the IMFs of a white noise (red curves) whose standard deviation has been adjusted to fit the SLA background noise between 30 and 15 km wavelength. It is presented for the Agulhas current area, top panel, and for the Gulf Stream area, bottom panel. For clarity, only the first three IMFs are shown. As expected, for white noise, the EMD filter bank is composed of a high-pass filter with the IMF1, and a low-pass filter bank with the higher ranked IMFs. A similar structure is observed for the IMFs of the SLA signal with identical cut-off wavelengths, which is the result of the noise shaping the frequency content of the SLA signal. This similarity shows the consistency of separate denoising of each IMF of rank n > 1 using the estimated noise variance given by Eq. (2). The IMF1 PSDs of SLA and white noise have a similar shape, both containing mostly high-frequency noise, but with higher SLA PSD values at scales > ~ 20 km, which is a consequence of the large modulation of the SLA noise by the varying sea state conditions and the inclusion of geophysical information such as that related to very large SLA gradients showing high SNR. These higher PSD values are even more important in the Gulf Stream area due to larger variety of sea states encountered and sampled by the altimeter passages.

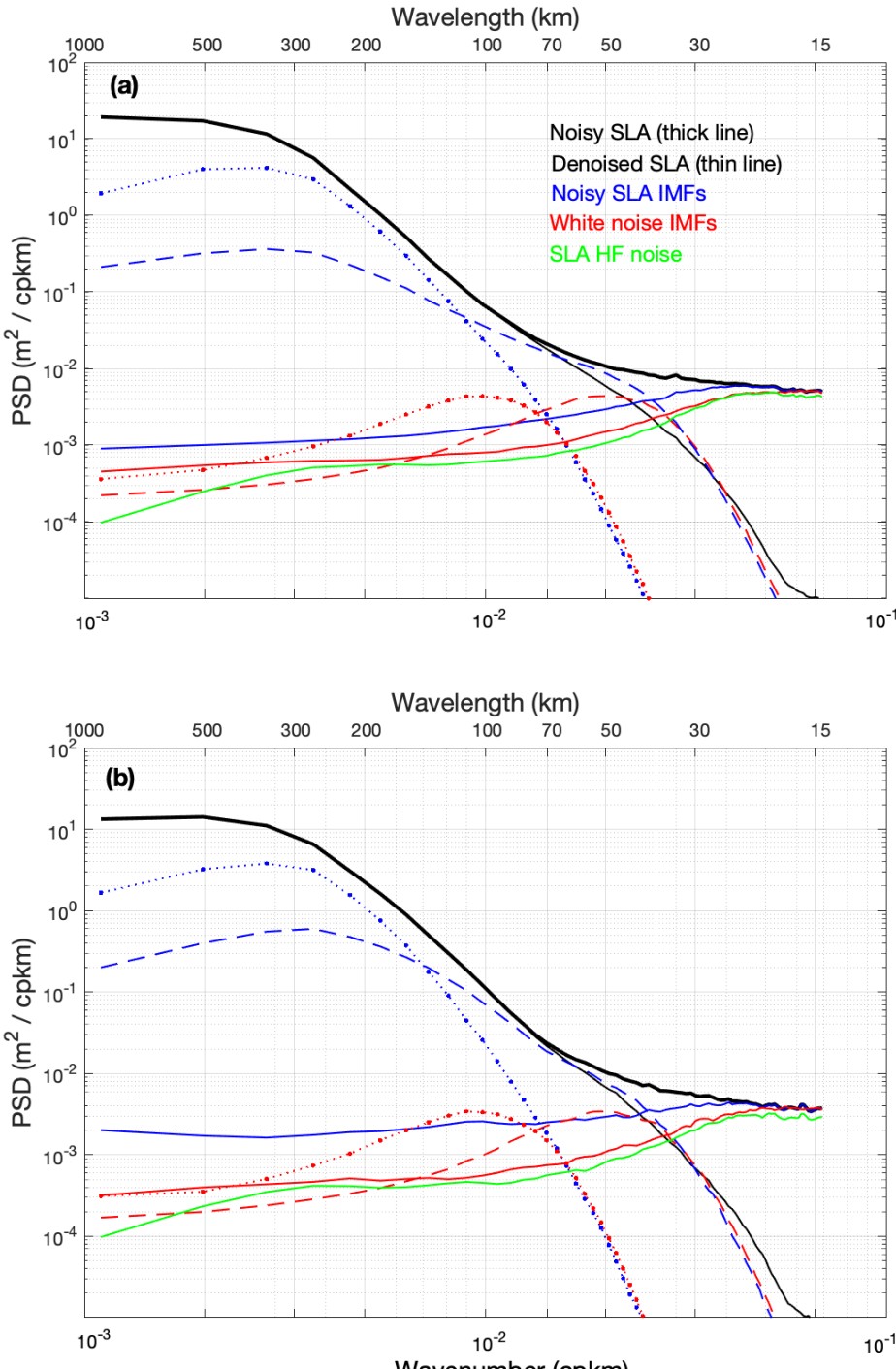

Figure 2: Mean Power Spectral Density (PSD) of the first 3 IMFs (1st=solid; 2nd=dashed; 3rd=dotted) for white noise (red curves) and SARAL SLA along-track measurements (blue curves), and mean PSD of the corresponding noisy (thick black line) and denoised (thin black line) SLA measurements. The PSD is the average of PSDs computed over all data segments covering the years 2016-2018, for the Agulhas region (10º W-35º W; 33º S-45º S, (a)) and the Gulf Stream (72º W-60º W; 44º-32º N, (b)) regions. The green line is for the PSD of the SLA high-frequency noise estimated from the SLA's IMF1 (solid blue line).

A is an important factor to adjust because it is directly related to the improvement obtained in the SNR. Kopsinis and McLaughin (2009) perform the optimization of the A factor by simulating a variety of input signals and SNR values. Following these results, first approximate values were tested for the altimeter data set, with a careful analysis of the obtained denoised SLA measurements. However, the adjustment of A for the different altimeters was refined. Indeed, the values of A showing the best SNR improvement on average may depend on the average SNR of the input signal, which is different for each altimeter. Sec. 4.2 details the practical rule for determining A that uses an approach to make the denoised SLA PSD consistent with the mean PSD of the unfiltered data minus the PSD of white Gaussian noise estimated in the range 15-30 km, and consistent for different altimeters.

## 4 Results

### 4.1 Examples

The two cases, Fig. 1, show AltiKa SLA measurements in the Gulf Stream area and Fig. 3 illustrates the EMD denoising principles for these examples. Described in Sect. 3.2, denoising a segment of SLA data after an initial expansion into an IMF set is a two-step process: 1) a wavelet analysis of IMF1 to separate and evaluate the high-frequency part of the Gaussian noise and any geophysical information embedded in IMF1; 2) EMD denoising of a set of 20 realizations of reconstructed noisy SLA series to estimate a mean denoised SLA series and its uncertainty.

Pass 597, right panels, is associated with rather low sea state conditions with little variability, and IMF1 (black curve, second right panel) has little amplitude modulation, but rather large phase modulation due to the the high SNR in several portions of the segment (few alternance of minima and maxima). Because of this relatively large phase modulation, a significant portion of the IMF1 is identified in the first step as "useful signal" by the wavelet analysis. In the second step however, this residual IMF1 signal will be almost completely removed. Indeed, it is well below the SNR prescribed by using the high-frequency noise jointly derived from the IMF1 wavelet processing and the threshold values set with Eq. (A1), (2) and (3) (blue lines in Fig. 3). Only a small modulation between data records 80 and 90 therefore shows up in the SLA denoised signal. The third panel shows IMF2, and its associated threshold derived from the IMF1 threshold (i.e. Eq. (2)), which maps the large SLA gradient in the Gulf Stream and mesoscale features near 70 km wavelength with some eddies appearing well above the threshold and other smaller amplitude oscillations that will be cancelled in the second denoising step. Already shown in Fig. 2, the SNR increases rapidly for IMF2 compared to IMF1. In this case, the uncertainty attached to the denoised SLA is almost constant below 1 cm, as shown in the fourth panel.

AltiKa pass 53 crosses the Gulf Stream 19 days before, but this is a very different situation. Quite frequent, such a case corresponds to high and variable sea state conditions with abrupt changes in SWH, shown Fig. 1. Strong westerly continental winds were present for several days before the AltiKa passage, which turned to the northwest the day before. SWH was less than 2 m near the coast between records 80 and 120, then a first large increase occurred on the northern side of the Gulf Stream near record 60, and a second on its southern side to reach sea state conditions with SWH > 5 m. Unlike the first case , the IMF1 thus shows a large modulation in amplitude, and relatively small modulation in phase. Since the phase modulation of IMF1 is primarily high-frequency for

wavelet analysis, it is analyzed as noise, and the "useful signal" is almost zero everywhere except for the largest IMF1 values. Furthermore, because the threshold value computed from the estimated noise is also larger (than for the first case), the IMF1 "useful signal" is completely removed in the second EMD denoising step. This highlights the potential of the denoising approach to handle variable sea state conditions. Nevertheless, the IMF2 processing

(third panels of Fig. 3) shows that a modulation of the SLA is assessed significant near the beginning of the IMF2 record, which may appear to be associated with larger noise values correlated with the high SWH values. It will remain an outlier, however, and will be adequately associated with the largest uncertainty in this data segment. Indeed, the uncertainty calculated as the standard deviation of the set of denoised signals increases with sea state, doubling along this data segment. Overall, these two examples confirm that the proposed denoising process adapts

well to varying sea state conditions.

In cases where sigma0 blooms or rain events corrupt limited portions of a data segment, and for which the data editing step did not perform, the impact is more difficult to analyze. It will depend on the magnitude and length of the associated errors which can vary greatly. However, the proposed EMD denoising process is not a data editing process and the results are certainly still affected by some of the largest errors. It should benefit from improvements

in data editing procedures and retracking algorithms that will be used for future CMEMS products.

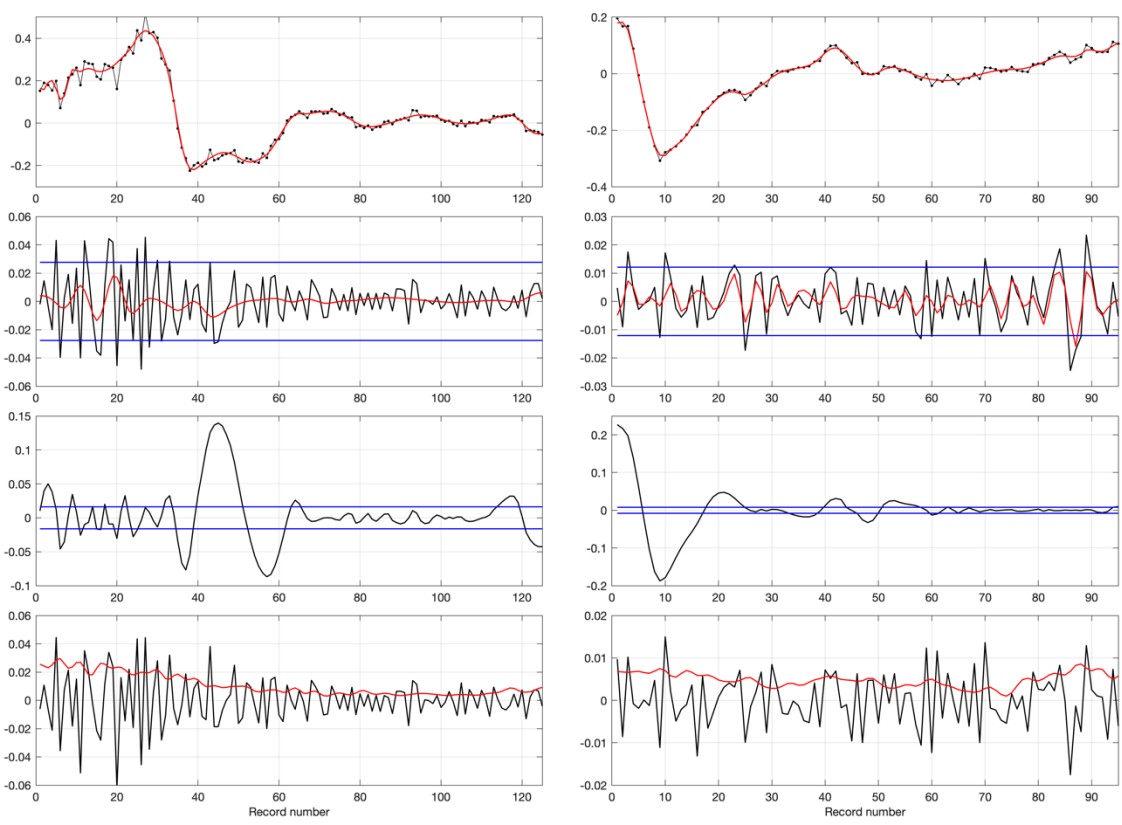

**Figure 3: SARAL short data segments in the Gulf Stream area for cycle 106 and passes 53 (left panels) and 597 (right panels). Top panels: noisy (black dotted) and denoised (red) SLA; Upper central panels: IMF1 (black), useful signal**

**(red) retrieved from wavelet denoising of IMF1, and thresholds (blue) to be applied in IMF1 EMD denoising; bottom central panels: IMF2 (black) and thresholds (blue) to be applied in IMF2 EMD denoising; bottom panels: noise (black)**

**retrieved from IMF1 wavelet denoising and uncertainty (red) attached to the denoised SLA. All units in meters, y-axis, and x-axis is the data record number.**


**4.2 EMD denoising calibration by PSD adjustment in the 30-100 km wavelength band**

For SLA measurements performed by a given altimeter instrument, the mean SNR is expected to vary primarily with sea-state. The mean SNR is then a function of the climatological distribution of sea-state conditions that are

dependent on ocean basins and seasons. The proposed denoising approach can efficiently adapt to the local SNR, allowing for a single global value for the control constant A in Eq. (3). However, since the noise statistics vary greatly with the average sea-state conditions, it is useful to show how such variability can impact the results when a single value of A is used in the global SLA processing. A two-step sensitivity study is performed below, which first determines specific A values for different regions, and then shows how the use of an overall A value impacts

the results.   A few areas are defined corresponding to different climatological sea-state conditions, namely the Gulf Stream region, the Agulhas current region, an area in the southern Indian Ocean, and an area in the central Pacific Ocean. The precise coordinates of the regions are given in the Fig. 4 legend. The analysis was then performed using AltiKa measurements, as the AltiKa PSD curve shows a well-defined expected white noise plateau in the high-frequency range, 15-25 km, shown in Fig. 4, which is not the case for Jason-3 and Sentinel-3.

We see that the height of the noise plateau depends on the climatological sea state conditions, with the highest value for the southern Indian Ocean, followed by the Agulhas, the Gulf Stream and the central Pacific Ocean. This is the effect of the dependence of the instrumental noise, after retracking, on significant wave height since the hump artifact does not depend on wave height (Dibarboure et al., 2014).

The two-step analysis for each region then follows:

1)   For each region, a set of discrete A values within a range corresponding to that found in Kopsinis and McLaughin (2009) are used to process the three-year data  set and obtain an average PSD of denoised measurements for each discrete A value. An optimal A value, for each region, is then found as the one that gives the best fit between the observed mean PSD and a mean SLA PSD calculated as the sum, over the data segments covering three years, of the denoised SLAs and a White Gaussian Noise (WGN) whose

average standard deviation is calculated to fit the mean observed PSD in the 15-25 km range. The best fit between PSDs is estimated as the root mean-square difference (RMSD) in the 30-100 km range. A minimum value for the RMSD is then found in the prescribed range of A. The denoised SLA PSD corresponding to this optimal value of A is the theoretical PSD that gives the best fit with the observed PSD for the white noise observed in the 15-25 km range, and is referred to as the "Best fit" PSD in Fig.

4. The "Best fit + WGN" PSD is in excellent agreement with the observed PSD in three of the regions. In the central equatorial Pacific, the observed difference may be related to the so-called hump artifact that can cause a deviation of the total noise PSD from a power law in $k^0$. This artifact is caused by backscatter inhomogeneities in the altimeter footprint associated with sigma0 blooms, rain cells (Dibarboure et al., 2014) and indeed contaminates altimeter measurements much more frequently in tropical regions.

Conflicting results are discussed in Dibarboure et al. (2014) regarding the spectral shape of the hump artifact, showing that it can be distributed as white noise or as a dome-shaped figure depending on the

analysis approach. In practice, it can also depend on the data editing, on the waveforms retracking, and on the way the PSDs are computed. Dibarboure et al. (2014) showed that a flat hump PSD is found when a large amount of long data segments are used to compute the PSD, which is not the case in the tropical oceans where hump artifacts are frequently edited, then reducing the length of continuous data segments. The PSD resulting from the classical analysis (Fu, 1983; Le Traon et al., 2008; Dufau et al., 2016) performed to estimate the SLA spectral slopes is also shown in Fig. 4 (referred to as "Observed – WGN" PSD). It is in good agreement down to 50 km with the "fitted PSD" for the three regions for which the "Best fit + WGN" PSD also agrees well with the observed PSD. The differences below 50 km wavelength may be due to the fact that the hump artifact resulting from contamination of portions of a number of data segments is not well distributed as white noise for our data segment collection.

2) An optimal value of A is therefore obtained for each region, ranging from 1.8 to 2.2. To show the results obtained when the same value of A is used for all regions, the same value of 1.8 was used to EMD-process simulated data calculated as the sum of the denoised SLA data corresponding to the theoretical PSDs of step 1 for each region ("Best fit" PSD in Fig. 4) and a white noise estimated in the 15-25 km range. The PSDs obtained by processing these simulated data with this single value of A are called "Retrieved" PSDs in Fig. 4 and are very close to the "Best fit" PSDs in all regions. This indicates that the process as a whole can provide a set of denoised measurements with a realistic energy distribution in the observed wavelength range. Further, any variation in the prescribed thresholds, which depend on the setting of A, is accounted for in the uncertainty parameter attached to the denoised measurements. Another way to assess the choice of the setting of the parameter A is to perform a sensitivity study using Monte-Carlo simulations. One thousand white Gaussian noise series and associated IMFs were generated. The IMF1s contain the high-frequency component of the noise series, whose spectrum is represented by the red solid line in Fig. 2. For each of the 1000 IMF1 series, the threshold for the EMD denoising process was computed using Eq. (A1), (2) and (3), and applied to test the IMF1s against the expected noise. The results show that more than 98.5%, 99% and 99.5% of the IMF1 data values are below the threshold with A values of 1.8, 2 and 2.2, respectively, which is the range of best fit values found in step 1 above for the different regions. EMD denoising is therefore effective in cases for which the additive noise is close to Gaussian, and is not very sensitive to variations found in A for the different regions. IMF1 values above the prescribed threshold can therefore be associated with geophysical information with good statistical confidence, especially since their significance will be tested further since denoising is achieved with an ensemble average of noisy processes. To process AltiKa globally, A is set to 1.925 which is the exact average value found for the four regions analyzed.

The problem to consider further is the presence, in limited portions of the processed data segments, of outliers associated with high waves or artifacts caused by sigma0 blooms or rain events. These will likely appear in IMF1 and IMF2 series and the most energetic events will not be thresholded since the thresholds are calculated using the median absolute deviation from zero of IMF1. For this reason, processing IMF1 using wavelet analysis is an important step to separate, as much as possible, the possible useful geophysical signal in IMF1 from outliers, and to estimate the underlying Gaussian noise.

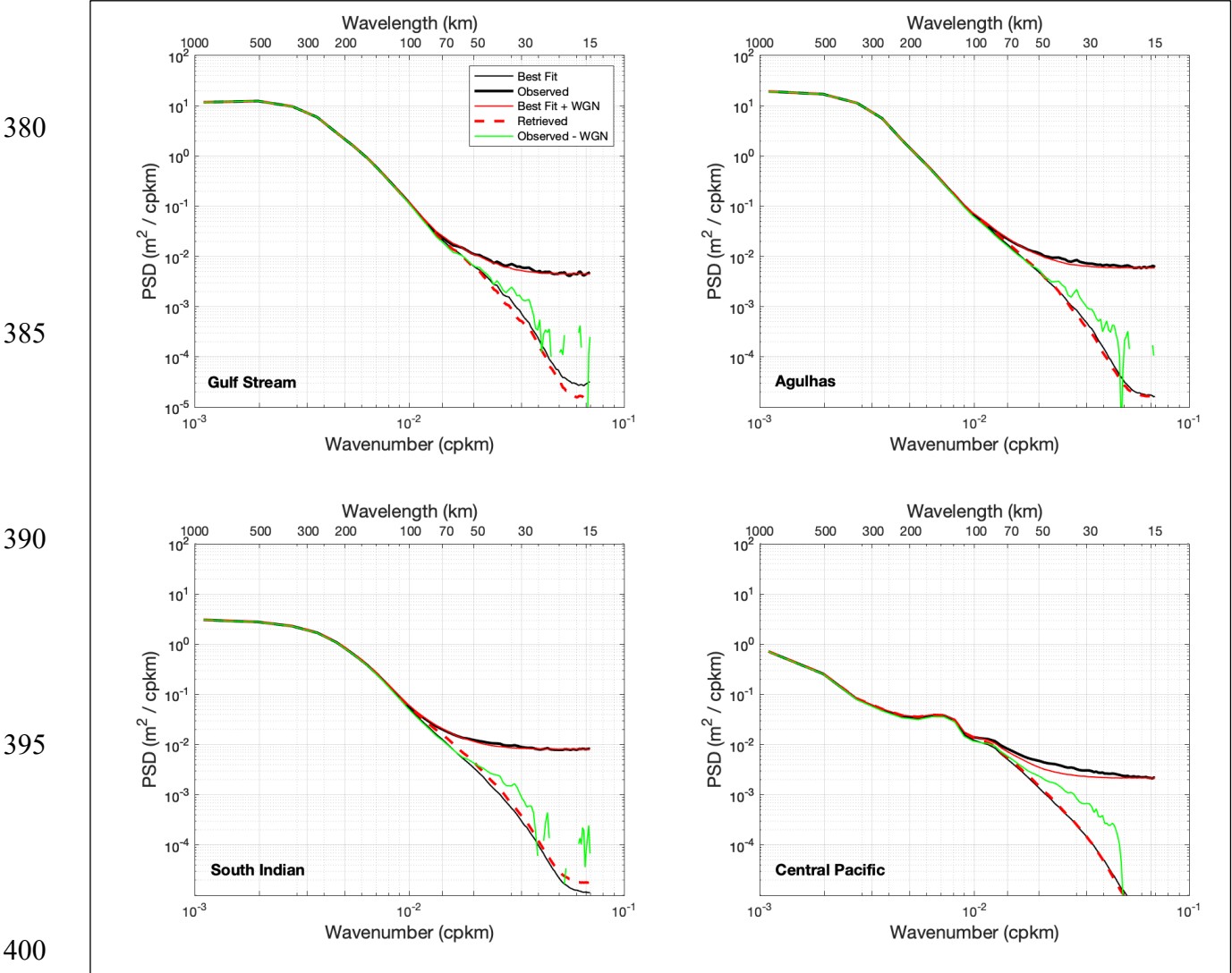

**Figure 4: Mean Power Spectral Density (PSD) of Saral SLA along-track measurements: Observed (thick black), Best Fit (thin black), Best Fit plus White Gaussian Noise (red), Retrieved (dashed red), Observed minus WGN (green). WGN is estimated as the average in the range 15-25 km of the mean observed PSD (bold black curve). The PSD is the average of PSDs computed over all data segments covering the years 2016-2018, and the Gulf Stream region (72º W-60º W; 44º N-32º N).**

The reasoning used above to set the control parameter A is based on the assumption that the measurements are affected by additive Gaussian noise with a known wavenumber dependence (in this case in $k^0$, as in studies correcting mean observed PSDs from mean white noise) , whereas this may not be the case in many along-track data segments where  artifacts related to sigma0 blooms, rain events, or  high sea state conditions may produce deviations of the noise from a Gaussian process (as shown in Fig. 1, left panels). Depending on the region, the "Observed PSDs" shown in Fig. 4 may be strongly shaped by these events, and this may explain the observed differences between the "Best fit" and "Observed – WGN" PSDs in Fig. 4, thin black and green curves

respectively. To further verify the consistency of the EMD denoising and to show the role of IMF1 processing, the 3-year dataset of denoised SLAs in the Gulf Stream region (PSD shown in upper left panel of Fig. 4) was considered the true geophysical signal of the SLAs, thus prescribing a theoretical PSD. For each data segment of these "theoretical" SLAs, a white noise of 1.8 cm standard deviation was added to provide a simulated noisy SLA segment. This data set was then processed with the EMD denoising algorithm. Figure 5 shows that the theoretical, EMD denoised, and "Noisy – WGN" PSDs are in excellent agreement over the entire wavelength range, in contrast to what is obtained with the observed SLA shown in Fig. 4. The same coherence is obtained regardless of the region analyzed. This means that the EMD denoising process is fully consistent in the case of measurements contaminated by a Gaussian noise, which also suggests that the hump artifact may deviate more or less from a white noise distribution for our data set.

In this simulation, the SNR is close to 1 on average near 50 km wavelength as shown in Fig. 5, but can be greater than 1 locally in a wavelength range down to 30 km. Such small mesoscale geophysical information emerging from the noise level can be retrieved from IMF1 using the dedicated wavelet denoising analysis. As shown in Fig. 5, the average PSD of IMF1 shows the plateau of high-frequency noise, but also significant energy content over a wider wavelength range associated with both lower frequency noise and geophysical information. The PSD curves of the IMF1 and the simulated SLA intersect between 50 km and 30 km wavelength. After wavelet decomposition of the IMF1, the wavelet denoising scheme specifies the maximum level to be retained for geophysical signal recovery. In the general case, only the level containing the finest scales is systematically discarded, and Fig. 5 shows the PSD of the signal recovered from IMF1 after using the Huang and Cressie (2000) denoising scheme (red dashed curve). The wavelet denoising acts as a low-pass filter with a sharp cut-off near 25 km wavelength and a significant amount of noise is also filtered out at longer wavelengths. In this simulation, the processing results in the recovery (red curve) of the full PSD (thin black curve) of the simulated signal because the A parameter has been set to do so (A=1.65), although in practice, and even though the SNR was dramatically improved (more than 97% of the IMF1 noise canceled), the geophysical signals with the lowest SNR were also filtered out. In this sense, and as expected, the SLA signal for the real data will only be partially resolved in the small mesoscale range. In specific cases, further filtering can be applied by setting a different maximum level for wavelet denoising of the real data, as outliers can contribute strongly to the IMF1 in the wavelength range 10-50 km. For example, a practical rule can be considered by further constraining IMF1 denoising when more than a given percentage of a processed data segment is associated with high seas. For outliers associated with sigma0 blooms and rain events, there is no simple relation between the noise associated with the hump artifact and sigma0 that would allow for such a practical rule. EMD denoising of real SLA measurements is therefore likely still contaminated by outliers in limited portions of a number of along-track data segments, and future improvements will depend on better data editing and implementation of the latest retracking algorithms (Passaro et al., 2014; Thibaut et al., 2017; Moreau et al., 2021).

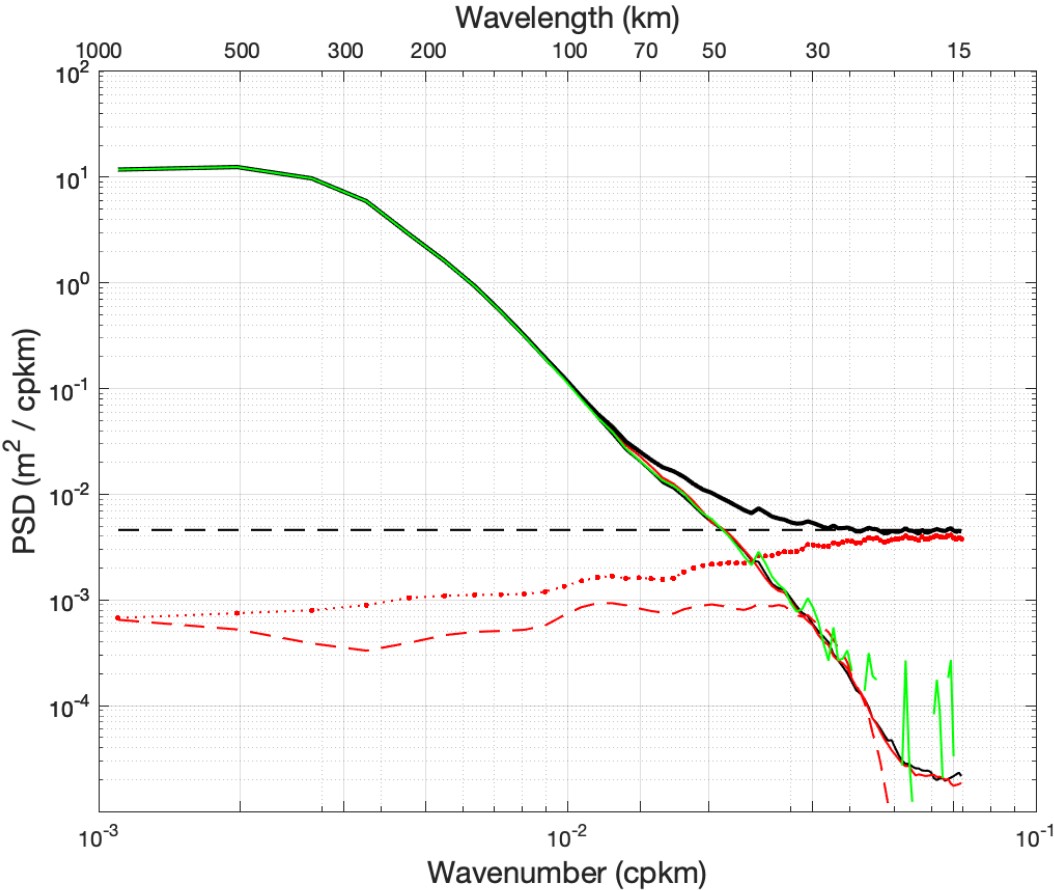


**Figure 5: Mean Power Spectral Density (PSD) of Saral SLA along-track measurements: simulated noise-free SLAs (thin black), simulated noise-free SLAs + simulated WGN with 1.8 cm std (thick black), EMD-denoised SLAs (solid red), IMF1 of SLAs (dotted red), noise-free portion of IMF1 obtained after IMF1 wavelet processing (dashed red). The above PSDs are computed as the average of PSDs obtained for all individual data segments covering the years 2016-2018, and**

**the Gulf Stream region (72º W-60º W; 44º N-32º N). The PSD shown as a green line is obtained by subtracting from the thick black line the mean WGN (std=1.8 cm) shown as the black dashed line (averaged over 15-30 km).**

### 4.3 Building a multi-sensor data set with consistent Power Spectral Density

The EMD denoising algorithm is then found to be robust and consistent in processing the AltiKa measurements. A workable rule can be defined to adjust the method to provide a global data set of denoised SLA measurements whose PSD are regionally consistent with the expected SLA geophysical signals. Such an approach is not easily applicable or numerically consistent for the Sentinel-3 and Jason-3 measurements. Their PSDs do not exhibit the expected white noise plateau in the 10-25 km wavelength range, Fig. 6. The red-type noise in Sentinel-3

measurements has already been discussed and analyzed in several studies, and has been shown to be mainly related to the effects of swell on SARM observations (Moreau et al., 2018; Rieu et al., 2021). The reason why the Jason-3 PSD also shows a tilted PSD in the high-frequency range is more puzzling. One possible explanation is that it results from poor data editing (especially the rain flag) for Jason-3. Indeed, while an effective rain flag was used for AltiKa so that rain has little influence on data quality (Verron et al., 2021), this is not the case for Jason-3,

which is therefore likely to be more impacted by rain events. Associated errors may shape the noise distribution

differently than white noise, as discussed in the previous section. Indeed, we found many more short segments of continuous measurements in the AltiKa data set than in the Jason-3 data set, both of which are distributed in the same CMEMS product, due to more efficient data editing. Therefore, the adjustment of the EMD denoising process for Jason-3 and Sentinel-3 was performed by using the AltiKa results as reference.

For Sentinel-3, Fig. 6 shows that its PSD for all analyzed regions is in excellent agreement with AltiKa's PSD over the entire wavelength range down to 25 km, which is a striking result showing that the two altimeters have similar average noise level and shape above 25 km wavelength. The same value of A was therefore used to set the noise thresholds for Sentinel-3, and it yields noise-free measurements PSDs in near perfect agreement with AltiKa. Note that, unlike AltiKa, the Sentinel-3 measurements are not sensitive to the hump artifact, thanks to the SAR

processing, and this difference is not apparent in the overall results shown in Fig. 6. Although the PSD shape of the Sentinel-3 noise has often been referred to as red noise, there are no published results showing that this is the case in the range of interest, i.e. wavelengths > 30 km. The similarity to the AltiKa PSD in this range might indicate that this is not the case, and this justifies the choice to retain the white noise configuration for the Sentinel-3 EMD processing. However, the processing is capable of dealing with other Gaussian noise figures, and the upper left

panel of Fig. 6 shows, for information, the result obtained for Gaussian noise with a Hurst exponent of 1, corresponding to a PSD in $k^{-1}$ represented by a green solid line in the 15-30 km wavelength range. The PSD associated with the denoised measurements is as expected slightly lower than the white noise case in the range 30-100 km.

For Jason-3 and the three-year data set analyzed, the control constant A was adjusted and set to 2.4 in order to

obtain the best fit of the PSDs of denoised measurements with AltiKa. Fig. 6 thus shows a high degree of consistency between the PSDs of the three altimeters, although each one only partially resolves the true geophysical content in the 30-100 km range, with Jason-3 doing worse than the other two. Indeed, the similarity of Jason-3 denoised PSD with the other two, while the noise level is significantly higher, indicates that the SNR has been less improved. However, this difference will be taken into account in the uncertainty parameter attached

locally to the denoised measurements, as documented in the next section. Note also that, although the EMD process adaptively accounts for the noisier Jason-3 measurements, a higher A threshold is necessary to obtain the best fit with the other altimeter PSDs. This results from the need to compensate for the poor data editing (especially rain flag) of the Jason-3 measurements, already discussed at the beginning of the section. Indeed, since the outliers affect only limited portions of an analyzed data segment, they do not affect the estimation of the EMD denoising

thresholds, which are calculated using the median absolute deviation from zero of the IMF1 noise. The thresholds are more tuned to the instrumental noise rather than to the total noise and outliers of large amplitude are not removed. They have a significant impact on the denoised PSD, so a larger A is needed to achieve similar PSD levels for Jason-3.

For reference, Fig. 6 shows a $k^{-4}$ law that indicates steeper slopes for the Gulf Stream and Agulhas regions, a slope

close to $k^{-4}$ in the south Indian region, and a flatter slope in the central Pacific. The variance in the 100-30 km range is the highest in the Gulf Stream region, followed by the Agulhas, south Indian and central Pacific regions, in descending order, in agreement with the results of Chen and Qiu (2021).

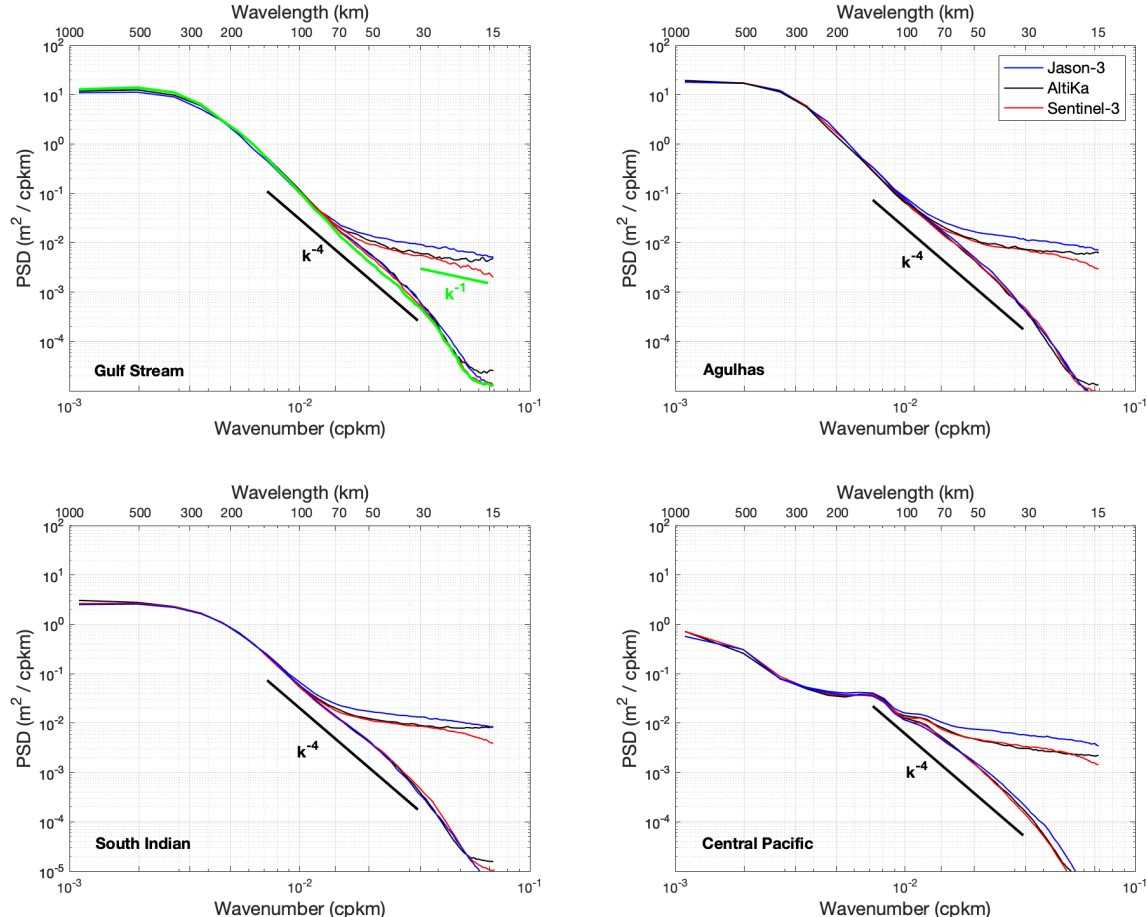

**Figure 6: Mean Power Spectral Density (PSD) of observed and denoised SLA along-track measurements for Jason-3 (blue), AltiKa (black), Sentinel-3 (red) and four regions: Gulf Stream (top left, 72º W-60º W; 44º N-32º N), Agulhas (top right, 10° E-35° E; 45° S-33° S); South Indian (bottom left, 80° E-110° E; 60° S-40° S); Central Pacific (bottom right, 170° W-150° W; 10° S-10° N). The PSD is the average of PSDs computed over all data segments covering the years 2016- 2018. For illustration, the green curve in upper left panel shows the PSD of denoised Sentinel-3 SLA when processed**
**with EMD and the hypothesis of a Gaussian noise following a $k^{-1}$ slope (pink noise, Hurst exponent = 1, shown as a green solid line in the range 15-30 km). The dark solid line shows a $k^{-4}$ slope in the range 30-150 km.**



## 4.4 Spectral slopes seasonality

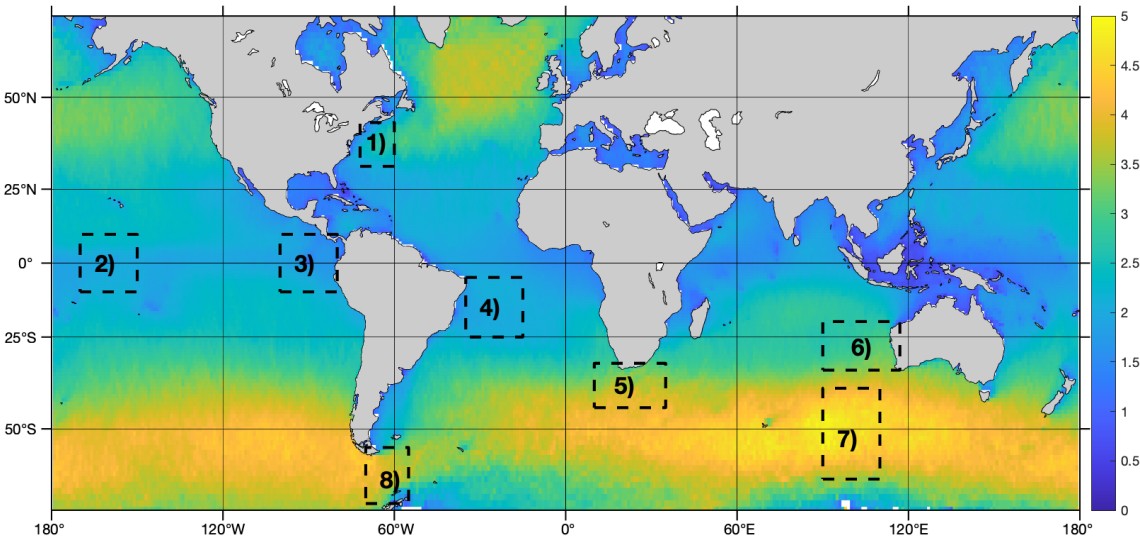


**Figure 7: Yearly-averaged significant wave height (m) computed over 2016-2018 from the Climate Change Initiative L4 products. Dashed black boxes define the 8 areas analyzed in the section.**

Seasonal variations in sea surface height in the small mesoscale range, with a wavelength less than about 100 km,
are very difficult to analyze because the noisy SLA spectral slopes are strongly shaped by errors related to the hump artifact, not dependent on sea-state, and by the instrumental and processing noise which is correlated with sea state conditions. Although the behavior of the resulting total noise is not well understood, it is genuinely postulated that the total SLA noise in the 1 Hz measurements can be considered white Gaussian when a large amount of long segments is used to calculate the spectrum. This enabled for the empirical study of seasonal
variations in SSH by removing an average noise PSD from the average PSDs of altimeter measurements (e.g. Vergara et al., 2019; Chen and Qiu, 2021). However, the applicability of this assumption may not be verified at the regional level, as suggested by the results presented in Fig. 4 and 5. This certainly also depends on the performance of the data editing. Therefore, the adoption of an alternative approach based on the analysis of the along-track EMD-denoised SLA measurements, rather than on the denoising of the SLA spectrum, is likely more
suited. It has also been shown that sea-state related errors are essentially removed by the EMD processing. Figure 6 shows that consistent spectral slopes of the denoised SLA are well obtained for the different altimeters. Hereafter, we only use AltiKa denoised measurements because the period covered is much longer for more consistent analysis of seasonal variations. The data used cover six years, from summer 2013 to winter 2018-2019, and the eight different regions shown in Fig. 7 are defined to cover various climatological sea-state conditions and expected
energy level in the small mesoscale variability of SLA.

For each region, the average SLA spectrum is shown in Fig. 8 for the boreal summer and winter, for the entire data set, and for a data set limited to segments having more than 80% of measurements with SWH < 4.5 m. This arbitrary SWH threshold correspond to the 90 percentile of the global data set and is intended to limit the influence of possible remaining outliers associated with extreme sea-state conditions. In regions of high climatological sea-
state, this thresholding will significantly limit the available data segments used to calculate the spectrum.

Distinct regions can be considered in Fig. 8. In the intratropical, regions 2 and 3 show no seasonality, a result in agreement with previous studies (Vergara et al., 2019; Chen and Qiu, 2021). In these regions, stable low sea-state conditions cannot introduce strong errors in the analyses. In the rough southern oceans, regions 7 and 8 (the Drake passage) show a small apparent increase in small mesoscale energy in the austral winter, disappearing when the high sea-state threshold is applied. In the latter case and for the Drake passage, 130 and 73 AltiKa passes satisfy the criterion and were used to estimate the mean spectrum for the boreal JJA and DJF, respectively. This suggests the absence of seasonality, in agreement with the results of Rocha et al. (2016) who used ADCP measurements in the Drake passage, and disagrees with Vergara et al. (2019) and Chen and Qiu ( 2021) who used the standard approach applied to altimeter data. Already mentioned, high sea conditions make it difficult to assess the results obtained by the different approaches, and better data editing and retracking algorithms (Passaro et al., 2014; Thibaut et al., 2017; Moreau et al., 2021) would improve the current analysis. The Agulhas region, number 5, which experiences mixed sea conditions, shows no seasonality, which is in agreement with the results of Chen and Qiu (2021). Three regions, 1, 4 and 6, show seasonality, insensitive to the filtering of high sea conditions. Strong seasonality in mesoscale dynamics on scales of 1-100 km, driven by turbulent scale interactions, are found in the Gulf Stream area using numerical modeling experiments and in-situ observations (Mensa et al., 2013; Callies et al., 2015), confirming the present altimetry results. Chen and Qiu (2021) also shows this strong seasonality in the Gulf Stream region and in region 6, west of Australia, as also obtained in our results. Conversely, we find seasonality in the western tropical Atlantic, region 4, not reported by Chen and Qiu (2021) study. Overall, for regions showing seasonality in the small 100-30 km mesoscale range that is apparently unaffected by high sea states events, SLA variability is found greater in winter of each hemisphere, consistent with stronger atmospheric being a source of enhanced submesoscale ocean dynamics (Mensa et al., 2013). For reference and evaluation of the data, and although different dynamics may be at work, a $k^{-4}$ spectral slope is shown in Fig. 8 for the 120-30 km wavelength range. It shows that, in the small mesoscale range, the steepest slope calculated from the PSD of denoised SLA measurements is found in the Gulf Stream region with a slope between $k^{-4}$ and $k^{-5}$. It is close to $k^{-4}$ in the Agulhas region and the high seas regions 7 and 8. In other regions related to the intratropics, flatter slopes are found, consistent with increased energy from internal tides and gravity waves (e.g. Garrett and Munk, 1972; Tchibilou et al., 2018). Overall, these different results are found to be consistent with studies using modeling experiments or observations from altimetry and in-situ data.

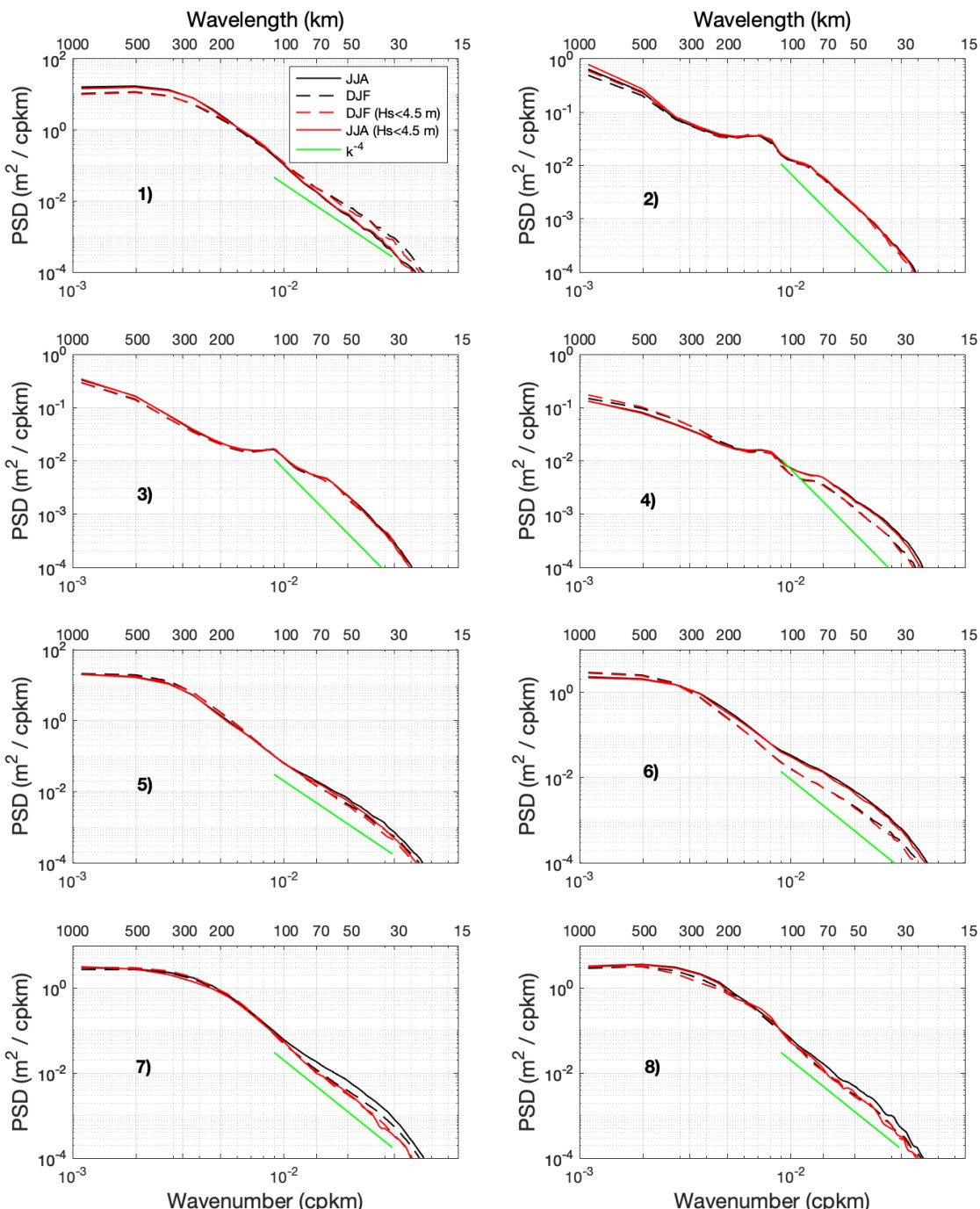

**Figure 8: Mean Power Spectral Density (PSD, m² cpkm⁻¹) of AltiKa denoised SLA in boreal summer (solid black curve, JJA) and in boreal winter (dashed black curve, DJF). The red curves show the results using a data set limited to segments showing more than 80% of data with SWH < 4.5 m. The numbered eight panels correspond to the 8 areas shown in Fig. 7. For reference, a PSD law in k⁻⁴ is shown in the wavelength range 120–30 km, green line.**

**4.5 Uncertainties in denoised sea level anomalies**

For a processed data segment, the resulting denoised SLA segment is the average of twenty realizations of the denoising process. An uncertainty $\varepsilon$, which characterizes the expected error attached to each denoised SLA in the data segment, is then calculated as the local standard deviation of the set of denoised SLA profiles corresponding to twenty random realizations of the noisy SLAs profiles. Since the noise process used to generate the set of random realizations is the high-frequency part of the observed Gaussian noise derived from IMF1, $\varepsilon$ readily accounts for the various errors affecting the data, i.e. instrumental and processing noise and remaining outliers, but it also represents the uncertainty related to the local SNR itself via the IMFs thresholding. The larger the local SNR, the higher the SLA modulation above the expected noise threshold, the lower the standard deviation $\varepsilon$, and vice-versa. The probability density function of IMF1, of the high-frequency noise, and of $\varepsilon$ is shown in Fig. 9. Sentinel-3 displays the best statistics, which is mainly a result of reduced errors related to the hump artifact in the SARM processing. Jason-3 has significantly higher levels of noise. These results are certainly strongly impacted by the different data editing performed for each instrument. It is however in agreement with other previous studies (e.g. Dufau et al., 2016; Vergara et al., 2019).

The uncertainty parameter $\varepsilon$ thus characterizes the error attached locally to each denoised SLA measurement, accounting for the variations in the signal to noise ratio but also reflecting the errors and choices made in the denoising process. Thus, the choice to impose similarity on the Jason-3, AltiKa and Sentinel-3 probability density functions, although the SNR of Jason-3 is significantly lower on average, implies less improvement in the SNR of Jason-3, resulting in worse statistics for $\varepsilon$.

The spatial distribution of $\varepsilon$ is shown in Fig. 10. It is mainly characterized by higher values in high-sea regions, in regions with high probability of rain events such as intertropical convergence zones, and also in regions for which the SLA variance is larger in the 30-120 km wavelength range, associated with a lower SNR. Interestingly, similarities are found with the altimeter 30-120 km sea surface height variance map analyzed by Chen and Qiu (2021). A more detailed analysis of the distribution of $\varepsilon$ is beyond the scope of this study but will certainly deserve further investigations.

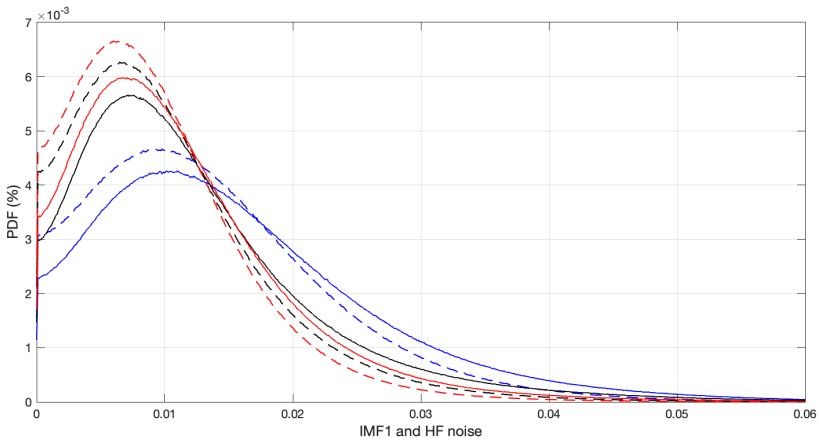

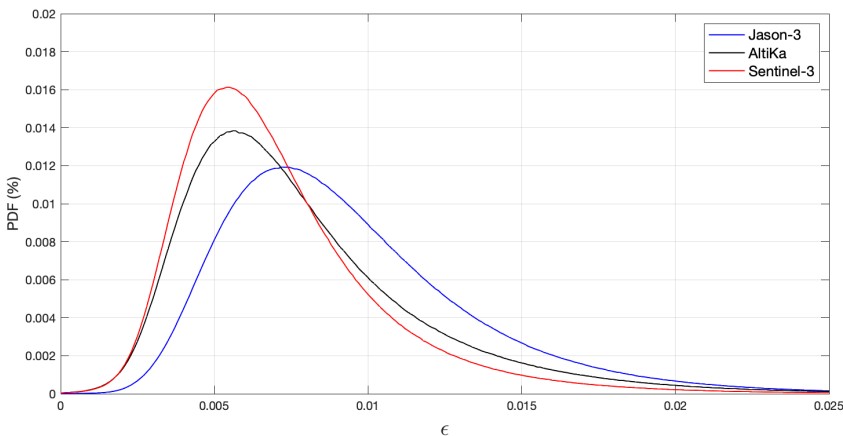

**Figure 9: Probability Density Function (PDF, %) of the absolute values of IMF1 (m, upper panel, solid lines), IMF1 HF noise (m, upper panel, dashed lines), and ε (m, bottom panel), for Jason-3 (blue), AltiKa (black), Sentinel-3 (red).**

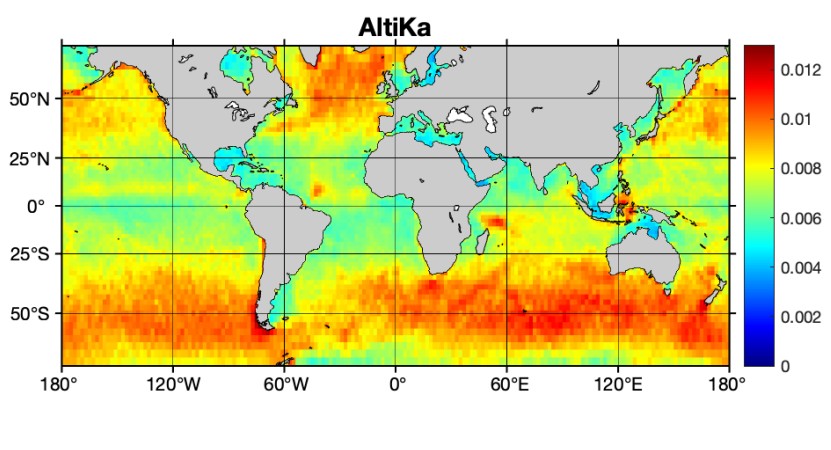

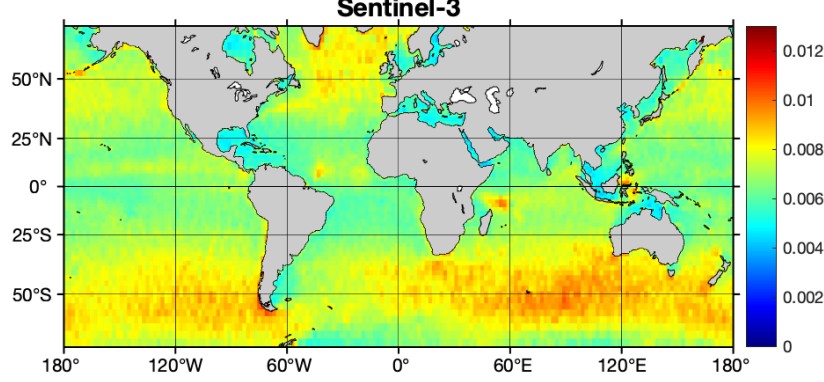

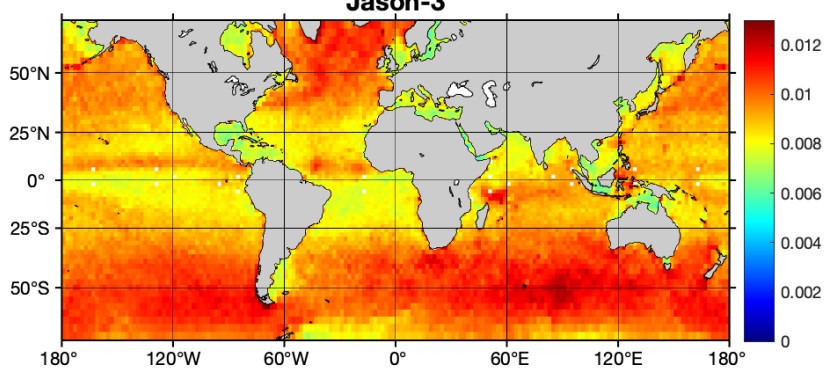

**Figure10: Three-year mean value of ε (m) for AltiKa (top), Sentinel-3 (centre), and Jason-3 (bottom).**

## 5 Key elements of comparison with other distributed products

We highlight in this study key features that make our approach to denoising SLA data different and more attractive than that currently used in other distributed products. It relies on: 1) the EMD algorithm that adaptively splits the SLA signal into a set of empirical functions that share the same basic properties as wavelets, but without having to conform to a particular mathematical framework; 2) a denoising algorithm that relies on a thorough and robust analysis of the local Gaussian noise affecting the SLA data over the entire wavenumber range; 3) an ensemble
average approach to estimate a robust denoised SLA signal and its associated uncertainty; 4) a calibration of the

method to provide a realistic distribution of SLA variability by adjusting the mean level of the power spectral density function.

It is therefore useful to compare our approach with the CMEMS products but also with the Data Unification and Altimeter Combination System (DUACS) experimental 5Hz products distributed by the Aviso+ center, as the latter
products include, among several differences from CMEMS processing, a high-frequency noise correction (Tran et al., 2019) that also aims to better retrieve mesoscale information in the 120-40 km wavelength range in preparation for the SWOT mission.  This comparison is limited to highlighting the key characteristic differences in the filtered products and a more in-depth comparison would be better addressed in a dedicated article.

For illustration, a selection of AltiKa passes in the Gulf Stream region is shown in Fig. 11. Panel (a) shows (same
pass as in Fig. 1 and Fig. 3, right panels) that EMD is best suited for analyzing strongly nonlinear signals to accurately map the large SLA gradient (more than 40 cm in less than 50 km), while CMEMS has the expected limitations/artifacts due to low-pass filtering, e.g. smoothing of gradients and poor localization of extrema. Panels (b) and (c) show two passes for which small mesoscale features (magnified in the insets) are recovered, and match well, for the SASSA and DUACS products, while the ~70 km cutoff applied in the CMEMS products suppresses
this information. Note that the SASSA denoising approach is based on a local SNR analysis and is therefore associated with a statistical estimate of the local uncertainty. In panel (c), the significant wave height is also displayed. It shows that the mesoscale variability in SLA, shown in the inset, does not appear to be associated with significant variability in SWH at 120-40 km scale, and thus may well be of geophysical origin, and not an artefact, in DUACS products, resulting from the high-frequency adjustment (HFA) correction. Indeed, the HFA correction
applied in DUACS products is based on a statistical relationship between the SLA and SWH retracking errors, at scales < ~120 km for which SWH variability is assumed to be only noise, in order to estimate the high-frequency SLA errors to be removed. Tran et al. (2019) showed that the HFA correction, associated with that of sea state bias, provides a 35% reduction in the noise variance affecting Jason-3 sea surface height measurements. Instead of panel (c) that shows a case for which the DUACS and SASSA denoised SLAs agree well, panel (d) shows a
common case where the DUACS result is exposed to contain more errors associated with the HFA correction. Indeed, a large variability of SWH at the < 120 km scale is observed in the vicinity of the Gulf Stream front, which is known to be the result of interactions between surface waves and current gradients. As a result, the Gulf Stream frontal system displayed in DUACS products is strongly offset from the SLAs observed by AltiKa, which does not seem correct. In these and many other cases (see left side of panel (d), the HFA correction likely induces errors
in the SLA signature, due to the wave/current interactions that shape the SWH field at scales down to a few kms (e.g. Kudryavtsev et al., 2017; Ardhuin et al., 2017; McWilliams, 2018; Quilfen et al., 2018; Quilfen and Chapron, 2019; Romero et al., 2020; Villas Bôas et al., 2020). One of the strengths of the EMD filtering approach used for SASSA products is that it does not rely on any assumption other than a Gaussian distribution of the noise contaminating SLA measurements.

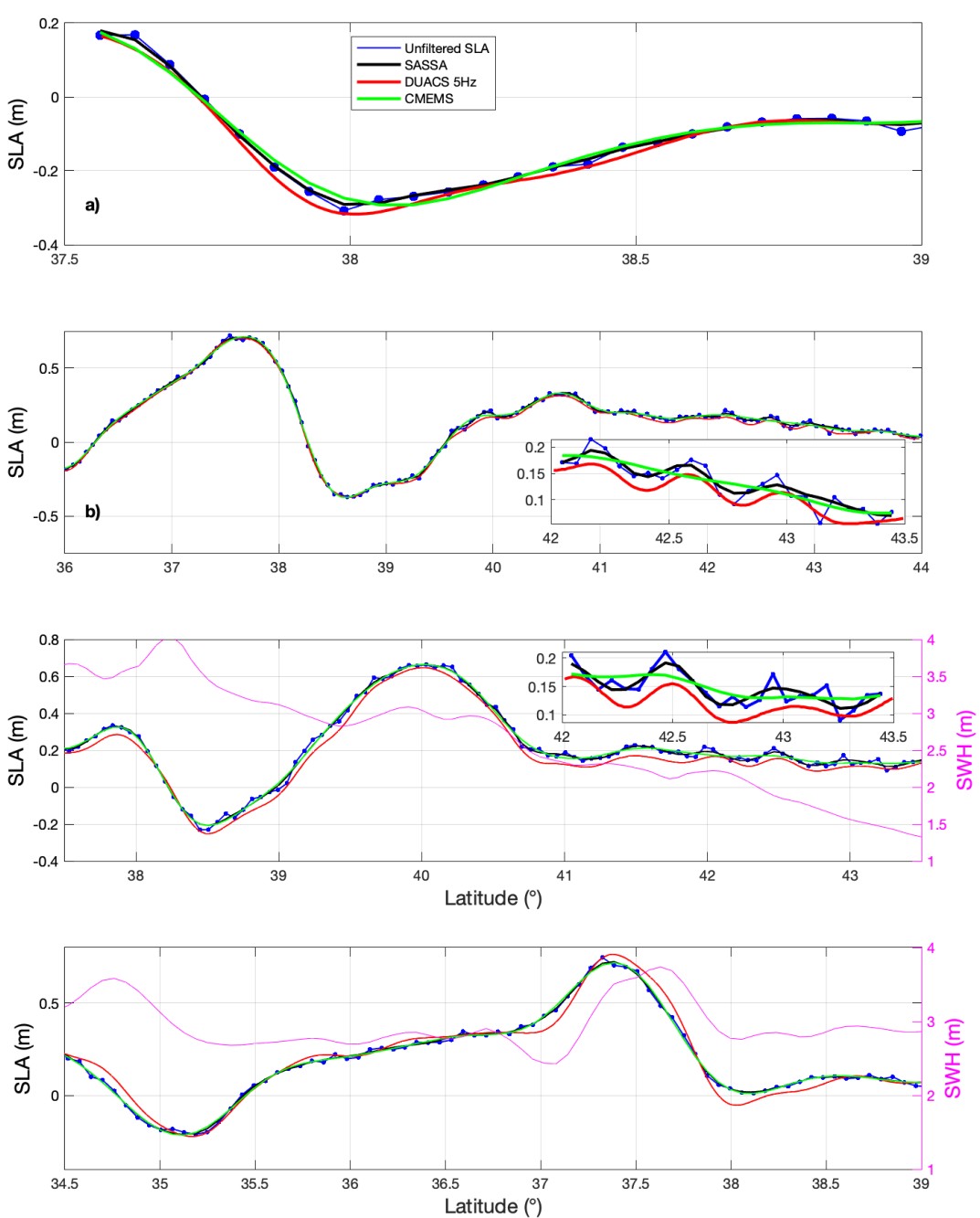


**Figure 11: CMEMS unfiltered (dotted blue), SASSA (black), DUACS 5Hz (red), and CMEMS filtered (green) Sea Level Anomalies (SLA, m) for different AltiKa passes: a) cycle 106 pass 597; b) cycle 101 pass 184; c) cycle 102 pass 941; d) cycle 103 pass 655. The magenta curve on right axis in panels c) and d) shows the SWH from the Sea State Climate Change Initiative products.**


Figure 12 shows the PSD for the same AltiKa products and the Gulf Stream region. For the CMEMS and DUACS products, the low-pass filter applied at about 65 km (CMEMS) and 40 km (DUACS) wavelengths results in a sharp decrease in PSD with increasing wavenumber, whereas the SASSA PSD is in close agreement with the PSD

obtained by removing white Gaussian noise (computed as the average PSD between 15 and 30 km wavelength) from the unfiltered SLAs. The fact that SASSA products can provide a "realistic / physical" representation of the SLA variance distribution over the entire resolved wavenumber spectrum is a direct result of the chosen approach. For DUACS products, it is unclear whether the variance between 40 km and 120 km wavelength corresponds to variance of unfiltered data or to variance of both unfiltered data and errors associated with HFA corrections

resulting from the geophysical variability of SWH at these scales, as discussed above.

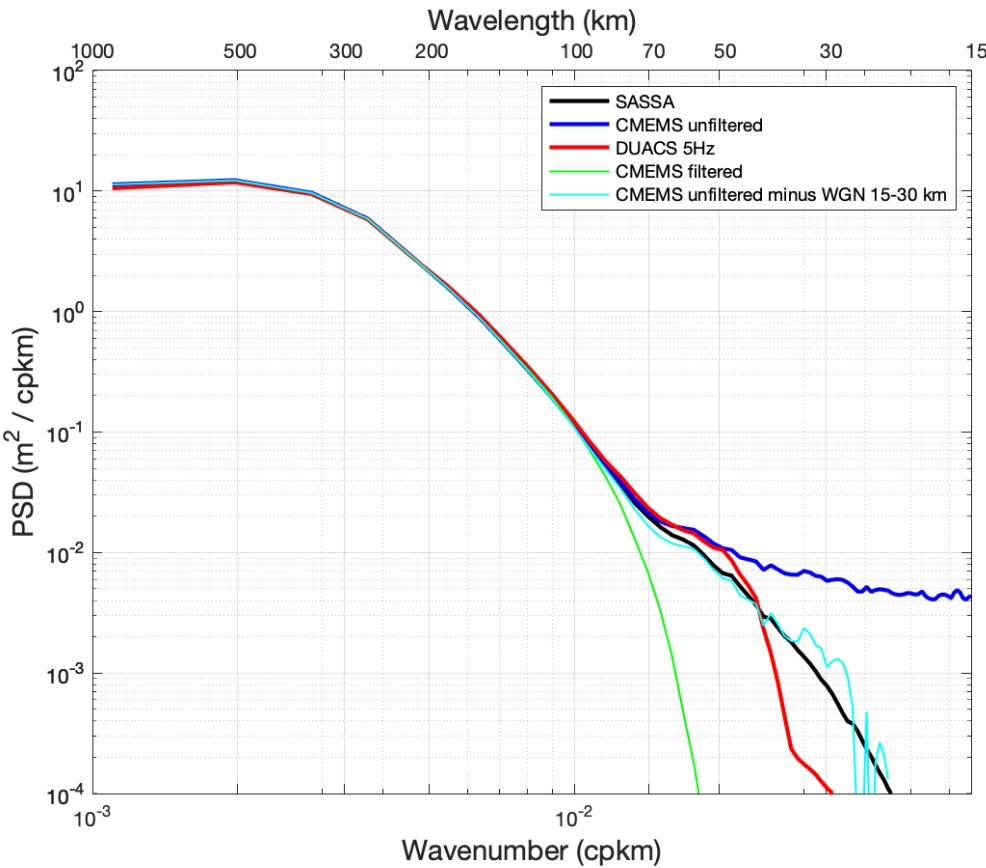

**Figure 12: Mean Power Spectral Density (PSD) of Saral SLA along-track measurements: CMEMS unfiltered (blue), SASSA (black), DUACS 5Hz (red), CMEMS filtered (green), CMEMS unfiltered minus an mean white gaussian noise (WGN) computed over 15-30 km wavelength. The PSDs are computed as the**

**average of PSDs obtained for all individual data segments covering the year 2017 and the Gulf Stream region (72º W-60º W; 44º N-32º N).**


## 6  Data availability

The SASSA dataset https://doi.org/10.12770/1126742b-a5da-4fe2-b687-e64d585e138c (Quilfen and Piolle, 2021) is freely available on the CERSAT website at ftp://ftp.ifremer.fr/ifremer/cersat/data/ocean-topography/sassa.  The altimetry observations used are obtained from the Copernicus Marine Environment Service and available at https://resources.marine.copernicus.eu/?option=com_csw&view=details&product_id=SEALEVEL_GLO_PHY_ L3_REP_OBSERVATIONS_008_062. Ancillary data are provided by the Sea State Climate Change Initiative products processed at the Institut Français de Recherche pour l'Exploitation de la Mer (IFREMER, Dodet et al., 2020) and available on the ESA CCI website at ftp://anon-ftp.ceda.ac.uk/neodc/esacci/sea_state/data/. The authors thank the reviewers for reviewing the manuscript and for providing feedback.

## 7 Code availability

The Matlab code used to generate the SASSA dataset is freely available on the CERSAT website at ftp://ftp.ifremer.fr/ifremer/cersat/data/ocean-topography/sassa.

## 8  Summary

Satellite altimetry is certainly ideally suited to statistically characterize ocean mesoscale variability thanks to its global, repeat and long-term sampling of the ocean. In particular, the estimation of sea surface height wavenumber spectra is a key unique contribution of satellite altimetry. However, below a wavelength of about 100 km, along-track altimeter measurements can be affected by a dramatic drop in the signal-to-noise ratio. It thus becomes very challenging to fully exploit altimeter observations for analysis of SLA distributions, especially within the small mesoscale 120-30 km range.

To overcome these difficulties, and to extend previous efforts to characterize spectral distributions, an adaptive noise removal approach for satellite altimeter sea level measurements is proposed. It essentially builds on the non-parametric Empirical Mode Decomposition method developed to analyze non-stationary and non-linear signals. It further exploit the fact that a Gaussian noise distribution becomes predictable after Empirical Mode Decomposition. Each altimeter segment can then be analyzed to adjust the filtering process.

Applied, this data-driven approach is found to consistently resolve the distribution of the SLA variability in the 30-120 km wavelength band. A practical uncertainty variable is then attached to the denoised SLA estimates that takes into account errors in the altimeter observations as well as uncertainties in the denoising process.

Here, measurements from the Jason-3, Sentinel-3 and Saral/AltiKa altimeters have been processed and analyzed, and their energy spectral and seasonal distributions more unambiguously characterized in the small mesoscale domain. In particular, the data-driven methodology helps to more consistently adjust to local sea state conditions. Anticipating data from the upcoming Surface Water and Ocean Topography mission (e.g. Morrow et al., 2019), these denoised SLA measurements for three reference altimeter missions already yield valuable opportunities to assess global small mesoscale kinetic energy distributions, as well as to study possible correlation between SLA high resolution measurements with sea state variability conditions.

**Appendix A: Denoising Scheme**

Data denoising is performed on data segments of 128 continuous measurements to limit large variations in noise statistics due to high sea state conditions. No gap filling is performed for missing values. In addition to the data editing performed for CMEMS products, additional outlier detection is performed to remove the largest isolated SLA peak values . For each data point in a segment, the difference in SLA with neighboring values is tested, within a sliding window of 5 points, and its SLA value is replaced by the average of neighboring values  if the difference is greater than 4.5 times the standard deviation of the IMF1 of the segment. For each data segment, a reference

high-frequency noise energy level,  $E_1$ , is estimated from the first IMF, to derive from Eq. (2) the expected noise energy in each IMF of rank greater than 1. $E_1$ is computed using the robust estimator given by the absolute median deviation (MAD) from zero, as follows:

$$E_1 = (median\,|n_1(t)|/0.6745)^2 \tag{A1}$$


where $n_1$ is an estimation of the noise contained in IMF1 rather than IMF1 itself. Indeed, although the MAD is expected to be robust in cases where the analyzed IMF1 signal contains residual values associated with a small amount of geophysical information or outliers (Mallat, 2009), it can nevertheless fail to be reliable in cases such as the one illustrated in the left panels of Figure 1. This is because Eq. (A1) assumes normality of the distribution,

and while it is close to be true in most cases,  a large variability in sea state conditions makes it less absolutely true.

The IMF1 processing to estimate the high-frequency Gaussian noise is however necessary and useful in two different parts of the denoising algorithm : 1) as discussed above, to compute $E_1$ verifying Eq. (A1) and then $E_n$ verifying Eq. (2) in order to compute the thresholds with Eq. (3) for denoising each IMF; 2) to obtain the high-

frequency noise series similar to that of the IMF1 of a white Gaussian noise, that will be used to generate a set of noisy signals and associated denoised signals whose average and standard deviation finally provides the denoised SLA signal and its associated uncertainty, respectively.

Figure 2 show the PSDs of the high-frequency noise estimated from the SLA IMF1 (green curves) which show very good agreement with the PSDs of the white noise IMF1s. The IMF1 processing has been tested using different

wavelet denoising algorithms available in the Matlab package, but the best result was obtained using the wavelet shrinkage method of Huang and Cressie (2000). These authors developed a Bayesian approach to denoising various signals, assuming that it can be the sum of the underlying targeted signal composed of a piecewise-smooth deterministic part and a stochastic part, plus a noise as a zero-mean stochastic part. The Symlet 8 function was used as the wavelet basis function.

To schematize, the EMD-based denoising algorithm is applied to each data segment as follows, with an iterative part of k iterations (actually 20) to perform the ensemble average process :

1) Perform an EMD expansion of the noisy signal x(t)

2) Perform the IMF1 wavelet denoising to separate the IMF1 stochastic noise, $n_1(t)$, from possible geophysical signal and outliers

3) Perform a reconstruction of signal xs(t) by adding the IMF1 geophysical signal with higher order IMFs

4) Randomly modify the positions of the noise $n_1(t)$ in successive windows of length k (about 120 km) to obtain a new noise realization $n_k(t)$ and a new noisy signal $x_b(t) = xs(t)+n_k(t)$

5) Perform an EMD expansion of $x_b(t)$

6) Carry out the denoising of IMFs by hard thresholding with Eq. (A1), (2) and (3), and reconstruct a denoised signal $x_k(t)$

7) Iterate k times steps 4 to 6 in order to obtain a set of denoised signals

8) Make an ensemble average of the $x_k(t)$ to obtain a robust denoised signal and an uncertainty estimate calculated as the standard deviation of $x_k(t)$

Noise removal for each IMF is done with a hard thresholding approach which is widely used in the field of wavelet denoising. The advantage of hard thresholding is that it preserves strong gradients (Kopsinis and McLaughin; 2009). Its adaptation to IMF thresholding is illustrated in Figure 5 of Quilfen and Chapron (2021). In practice, for each IMF of rank n, the modulation intervals that are below the prescribed threshold $T_n$, given by Eq. (3), are set to 0.

### Appendix B: Power Spectral Density calculation

The SLA wavenumber spectra are calculated in regions of different sizes using Fast Fourier Transforms (FFT), after detrending and applying a 50% cosine taper window (Tukey window), for overlapping ground track segments of 128 continuous measurements. This corresponds to segment lengths of about 800 km or more, which is adequate for our study for which we focus primarily on the wavelength range below 120 km. Each mean spectrum is computed as the average of the individual spectra over at least the 2016-2018 period common to all 3 altimeters, which ensures that a sufficiently large number of segments are used.

### Author contributions

YQ and BC conceived and organized the manuscript. YQ wrote the manuscript with inputs from all authors. YQ and JFP collected and processed the data. YQ implemented the EMD denoising method. JFP supervised the overall production of the SASSA dataset.

### Competing interests

The authors declare that they have no competing interests.

**Acknowledgments**

This study was conducted within the Ocean Surface Topography Science Team (OSTST) activities. A grant was awarded to the SASSA project by the TOSCA board in the framework of the CNES/EUMETSAT call CNES-DSP/OT 12-2118. Altimeter data were provided by the CMEMS at https://resources.marine.copernicus.eu/?option=com_csw&view=details&product_id=SEALEVEL_GLO_PHY_ L3_REP_OBSERVATIONS_008_062s, for 1Hz SLA data, by the CCI project at https://climate.esa.int/fr/odp/#/project/sea-state for 1Hz SWH data, and by SSALTO/DUACS (distributed by

AVISO+, https://www.aviso.altimetry.fr) with support from CNES, for 5Hz SLA data.

The authors are grateful to the two reviewers for their constructive comments.

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
