# Peer review of "Towards improved analysis of short mesoscale sea level signals from satellite altimetry"

_Earth System Science Data, 2021_

## Author Response (AR1)

**Author's responses to referees/public comments on essd-2021-352**

*In the text below, the referees/public comments are in regular type and our responses are in italics.*

**Responses to referees comments**

**1) Responses to referee#1 comments**

Dear all,

After revising the article "Towards improved analysis of short mesoscale sea level signals from satellite altimetry" (Authors: Yves Quilfen, Jean-François Piolle, and Bertrand Chapron), my opinion is that the manuscript can be published as it is.

I found this paper very relevant and complete.

Best regards,

*The authors thank the reviewer for agreeing to review the manuscript and for her positive criticism.*

*Best regards*

**2) Responses to referee#2 comments**

The authors provide a description of a new sea level anomaly dataset based on Altika, Jason3 and Sentinel3 along-track observations filtered using the EMD method. Overall I found the paper exhaustive and well written. EMD is a novel filtering method with lots of potential for geophysical application and the retrieval of mesoscale information from satellite altimetry an important topic for current as well as future missions. The manuscript represent a valid and important contribution to the field, but there are 25 some aspects that I would like to have better clarified before the manuscript is finally published.

*Thank you for reviewing our manuscript and for providing positive feedback. We are responding to the various points below, and the manuscript will be modified appropriately to address the reviewer's comments.*

**Major remarks**

I found that one of the main limitations of this paper is the lack of a good term of comparison against which to evaluate the performance of the EMD filtering. The authors remark several times how current filtering methods remove almost entirely the altimetry signal at scales below ~70 km and how for that reason EMD should be preferred. However, a comparison between the official filtered CMEMS products and the EMD ones is never presented. I think that the addition of a spectral comparison between the two as well as between across-track SLA signal for the two tracks showed in Fig 1 would further strengthen the author's conclusions.

*Originally, we intended to show the results of the comparison between EMD-denoised (SASSA) and official CMEMS products, as well as with the experimental SSALTO/DUACS 5Hz products, but we felt that this would have resulted in too long an article if an in-depth comparison had been performed. Given*

*the reviewer's comment and suggestion, we propose to add a short subsection before the "Data availability" section to provide an illustration of the salient features that make our approach different and more attractive. In the revised manuscript to be submitted after this interactive discussion part, the content of the new section will be related to what is discussed below. However, it should be clear that we cannot perform a thorough comparison of the different approaches, a task that should eventually be*

*the subject of a separate paper. As the data set is freely available, any interested scientist/user will eventually be able to make such a comparison.*

*It is useful to compare our approach with CMEMS products but also with the 5Hz experimental DUACS products proposed by Aviso, as the latter products include, among several differences from CMEMS processing, a high-frequency noise correction (Tran et al., 2019) that also aims to better retrieve*

*mesoscale information in the 120-40 km wavelength range. For illustration, a selection of AltiKa passes in the Gulf Stream region is shown below. Panel a) shows (same pass as in Figure 1 of the manuscript, right panels) that EMD is best suited for analyzing strongly nonlinear signals to accurately map the large SLA gradient (more than 40 cm in less than 50 km), while CMEMS has the expected limitations/artifacts due to low-pass filtering, e.g. smoothing of gradients and poor localization of*

*extrema. Panels b) and c) show two passes for which small mesoscale features (highlighted in the insets) are recovered, and match well, for the SASSA and DUACS products, while the ~70 km cutoff applied in the CMEMS products suppresses this information. Note that the SASSA result is based on a local signal-to-noise ratio (SNR) analysis and is associated with an uncertainty estimate. In panel c), the significant wave height (Hs, from Sea State CCI) is displayed. It shows that this mesoscale variability in SLA is not*

*associated with significant variability in Hs and thus may well be of geophysical origin, and not an artefact resulting from the HFA correction. Indeed, the HFA correction applied in DUACS products is based on a statistical relationship between the SLA and Hs retracking errors, at scales < ~120 km for which Hs variability is assumed to be only noise, in order to estimate the high-frequency SLA errors to be removed. Panel d) presents a common case where the DUACS result is exposed to contain many*

*more errors associated with the HFA correction. Indeed, a large variability of Hs at the < 100 km scale is observed in the vicinity of the Gulf Stream front, which is known to be the result of interactions between surface waves and current gradients (many studies show this). In these and many other cases (see left side of panel d), the HFA correction likely induces errors in the SLA signature, due to the wave/current interactions that shape the Hs field at scales down to a few kms.*

[Figure]

*Figure 1: CMEMS unfiltered (dotted blue), SASSA (black), DUACS 5Hz (red),  and CMEMS filtered (green) Sea Level Anomalies (SLA, m) for different AltiKa passes: a) cycle 106 pass 597; b) cycle 101 pass 184; c) cycle 102 pass 941; d) cycle 103 pass 655. The magenta curve on right axis in panels c) and d) shows the Hs from the Sea State Climate Change Initiative products.*

*Figure 2 shows the power spectral density (PSD) for the same products and the Gulf Stream region. For the CMEMS and DUACS products, the low-pass filter applied at about 65 km (CMEMS) and 40 km (DUACS) wavelengths results in a sharp decrease in PSD with increasing wavenumber, whereas the SASSA PSD is in close agreement with the PSD obtained by removing white Gaussian noise (computed as the average PSD between 15 and 30 km wavelength) from the unfiltered SLAs. The fact that SASSA products can provide a "realistic / physical" representation of the SLA variance distribution over the entire resolved wavenumber spectrum is a direct result of the chosen approach. For DUACS products, it is unclear whether the variance between 40 km and 120 km wavelength corresponds to variance of unfiltered data or to variance of both unfiltered data and errors associated with HFA corrections resulting from the geophysical variability of Hs at these scales, as discussed above.*

[Figure]

*Figure 2: Mean Power Spectral Density (PSD) of Saral SLA along-track measurements: CMEMS unfiltered (blue), SASSA (black), DUACS 5Hz (red), CMEMS filtered (green), CMEMS unfiltered minus an mean white gaussian noise (WGN) computed over 15-30 km wavelength. The PSDs are computed as the average of PSDs obtained for all individual data segments covering the year 2017 and the Gulf Stream region (72° W-60° W; 44° N-32° N).*

My second major remark, regards the Bayesian/wavelet filtering of the IMF1. It seems a quite
complex step to be included in the analysis, for which not too many details are provided (for
instance which wavelet base is used?) and it does not seem to give a big return. The examples
in Figure 4 both show that only a one peck of the IMF1 is retained as significant signal, while
the rest of the points are discarded. All the spectra shown indicate, that the IMF1 filtered energy
retained at longer wavelengths is only a very small percentage of the total reconstructed signal
(at least one order of magnitude smaller). Given the examples shown in figure 4 it is unclear to
me how the spectra shown in figure 6 can be obtained and what the associated filtered IMF1
would look like. While the examples shown in figure 4 are appropriate to explain the variations
in the signal and signal noise as a function of varying SWH conditions, they do very little to
convince the reader on why such IMF1 denoising step should be included in the analysis. The
author mention (lines 204-205) that "processing IMF1 using wavelet analysis is an important
step to separate, as much as possible, the possible useful geophysical signal in IMF1 from
outliers", but I'm not convinced by the examples they showed and I found much more
convincing their comments regarding the importance of data editing in those situations (e.g.
lines 122-125 or 480-485). Thus, I encourage the authors to provide a bit more evidence to
justify the inclusion of such complex step in an already fairly complex (and novel) filtering
method.

*Inspired by the work of Kopsinis and McLaughin (2009), based on numerical experiments, the
proposed EMD-denoising algorithm has been adapted to altimetry data for which the SNR
varies greatly and may have locally non-stationary noise statistics, and which are often affected
by outliers. Managing these non-friendly features as well as possible indeed introduces a
certain degree of complexity in the process (compared to the simple low-pass filtering applied
in CMEMS products), and the specific processing of IMF1 by wavelet analysis (or by some
other means) is indeed an important feature that brings a great return, with two main
objectives:*

1) *In cases where the SNR is locally high, at the sampling rate of the analyzed data, the
   sifting process used to derive IMFs likely results in the inclusion of geophysical
   information in the IMF1 because it is based on the detection of the two extrema
   envelopes (maxima and minima). The two examples in Figure 4 of the manuscript were
   not chosen to illustrate this feature. Instead, the examples in Figure 3 below are
   frequent and typical cases that show the need for IMF1 processing (as well as the two*

*examples showing insets in Figure 1 above). It shows a comparison between the simplest approach of  removing IMF1 from the unfiltered data to obtain a high-frequency noise-free signal, and the full denoising process. As can be seen, IMF1 is significantly modulated in amplitude and phase in regions with well-defined mesoscale variability, which may be geophysical in origin, or with large gradients. By removing the IMF1 entirely from the unfiltered SLAs, it is clear that features potentially of geophysical origin can be missing and that this degrades the representation of large gradients.  One of the main goals of this work is to be able to retrieve this small mesoscale variability and associate it with an estimate of the uncertainty. The reason it does not appear as a significant contribution in the averaged PSDs is that the PSDs are averaged over a large sample of data segments, not all of which showing such variability, which is furthermore often limited to small portions of the track. However, the examples below show the improved mapping of SLA variability when associated with sufficient SNR, and the better mapping of large gradients. Note that if one were to process AltiKa data from 5Hz waveforms, the IMF1 would certainly be all noise and the methodology adapted to that. There is indeed a great deal of flexibility in the method, which makes it, if not very simple, very adaptable.  However, as noted in the manuscript, better editing of the data before denoising step  will make it even more effective.*

[Figure]

*Figure 3: Three AltiKa passes; top panels: unfiltered (blue), unfiltered minus IMF1 (black), EMD-filtered (red) SLAs (m); bottom panels: SLA's IMF1 (red, m), denoised SLA's uncertainty (black, m)*

2) *The other main reason for processing IMF1 using wavelet analysis is to estimate the high-frequency noise series from the IMF1, that would correspond to the IMF1 of a Gaussian noise. The noise series is used in two ways to process a data segment. First, to estimate the variance of the IMF1 Gaussian noise, to calculate the noise variance of higher order IMFs and the derived threshold values. If the IMF1 contains too much geophysical information or outlier signature to significantly shape the IMF1 amplitude distribution, the noise variance estimated for all IMFs would be biased even though the Median Absolute Deviation is expected to be a robust estimator. Second, and perhaps more importantly, the noise series is used to generate a set of noisy series to obtain a robust estimate of the denoised SLA associated with an uncertainty value. This assumes that the noise is Gaussian. Figure 4 below, as well as Figures 2 and 3 in the manuscript, show that IMF1 denoising does a good job of testing this assumption since the PSD of the noise estimated from IMF1 and Huang and Cressie's wavelet denoising (blue curve) is very close to the PSD of IMF1 for white Gaussian noise (red curve), which is not the case for the SLA IMF1's PSD (dashed black curve). For information, several denoising schemes using wavelets have been tested with Matlab functions (green curves) and the one of Cressie and Huang (not included in Matlab) gave better results as shown in Figure 4. This has already been discussed in lines 233-243 of the manuscript. The wavelet basis used is Symlet 8. Different bases were tested, but this has much less impact on the results than the choice of wavelet denoising scheme. It is now informed in the new version of the manuscript, that includes an Appendix to detail the denoising scheme and explain why IMF1 denoising is necessary.*

[Figure]

*Figure 4: Mean Power Spectral Density (PSD) of IMF1 for white noise (red curve) and SARAL SLA along-track measurements (dashed black curve), and mean PSD of the corresponding noisy (thick black line) and denoised (thin black line) SLA measurements. The green and blue lines is for the PSD of the SLA high-frequency noise estimated from the SLA's IMF1 with different Matlab denoising schemes and*

*with the Huang and Cressie (2000) scheme, respectively . The PSD is the average of PSDs computed over all data segments covering the year 2017, and the Agulhas region (10° W-35° W; 33° S-45° S).*

Finally, the authors mention that Sentinel-3 provides observations in both SARM and LRM, describing the latter as "same processing as Jason-3 and AltiKa". That is incorrect and should be modified. The SRAL mission is always operated at High Resolution Mode (commonly called SAR mode). Low Resolution Mode (LRM) is a back-up mode only. (see https://sentinels.copernicus.eu/web/sentinel/user-guides/sentinel-3-altimetry/overview/modes ). 1Hz observations from Sentinel-3 are from the so called pseudo-low resolution mode (PLRM) which was designed to be as analogous as possible to the Jason-3 low-resolution processing, but it is not exactly the same (for instance less individual waveforms are averaged together in PLRM, so that it is characterized by slightly higher noise than Jason-3).

*Our description of the Sentinel-3 data may indeed be confusing. We have modified it as follows:*

*"The altimeter on board Sentinel-3 is a dual-frequency Ku-C altimeter that differs from conventional pulse-limited altimeter in that it operates in Delay Doppler mode, also known as Synthetic Aperture Radar Mode (SARM). SARM is the primary mode of operation which provides ~ 300 m resolution along the track."*

*However, it is our understanding that AVISO L2p products are used as input to Sentinel-3 1-Hz products provided by CMEMS and that "the sea level anomaly considered in Sentinel-3 L2P products is always based on Synthetic Aperture Radar (or if not available on Low Resolution Mode) data, but never on Pseudo LRM data" as stated in the L2p products Handbook. Please correct us if we are wrong.*

**Minor Remarks**

Since in Flandrin et al. (2004) alpha is defined as 2H-1, shouldn't its value in equation (1) be 0? This would make the error variance vary as $2^{-n}$ rather than being constant ($2^0=1$).

*Thank you for pointing out this typo in the alpha value, line 188. For white noise, H=0.5 alpha=0, which effectively gives the variance of noise to vary as $2^{-n}$.*

Legends should be included in Figures 3 and 6

*Done in the revised manuscript.*

Please explicitly indicate in the paragraph between lines 220 and 224 if any further editing or gap filling has been applied to the input data

*The following text will be provided in the revised version: 'Data denoising is performed on data segments of 128 continuous measurements to limit large variations in noise statistics due to*

*high sea state conditions. No gap filling is performed for missing values. In addition to the data editing performed for CMEMS products, additional outlier detection is performed to remove the remaining isolated peak values for the SLA. For each data point in a segment, the difference in SLA with neighboring values is tested, within a sliding window of 5 points, and its SLA value is replaced by the average of neighboring values if the difference is greater than 4.5 times the*

*standard deviation of the IMF1 of the segment.'*

Section 4.2 could be improved : the initial impression is that the parameter "A" should be defined according to seasons/region, but then the authors show that this is not the case and a mean value specific for each satellite mission should be used. This sort of pulls the rug under the reader feet. I think it would be easier for the reader if this conclusion was introduced at the beginning of the section to better guide him through the text.

*We describe in the introductory part of section 4.2 how the value of parameter A is defined, see lines 345-357. To address the reviewer's concern, we have modified the text as follows:*

*"The proposed denoising approach can efficiently adapt to the local SNR, allowing a single*

*global value for the control constant A in "Eq. (4)". However, since the noise statistics vary greatly with the average sea-state conditions, it is useful to show how such variability can impact the results when a single value of A is used in the global SLA processing. A two-step sensitivity study is performed below, which first determines specific A values for different regions, and then shows how the use of an overalll A value impacts the results."*

Lines 360 to 375: Please explain the meaning of the WGN acronym in the text. Furthermore, text and figure 5 legend are not consistent ("Best-fit" and "Fitted-PSD"). Please correct that.

*The acronym White Gaussian Noise (WGN) is now defined in the text and the inconsistency between the text and the Figure 5 legend corrected.*

Why are the green PSD curves in figure 5 not continuous but show gaps at small wavelengths?

*This is because the average noise PSD is calculated as the average PSD between 15 and 30 km, negative values are obtained at some wavenumbers in this range when removed from the actual PSD values. These values are therefore ignored when plotting the logarithm .*

Lines 471-472: shouldn't that be red dotted and red dashed lines, instead of "black and red dashed"?

*Indeed, this is not correct and not so simple. In fact, the total noise in IMFs of rank > 1 should be: dashed black – (dotted red – dashed red). The sentence has been removed because it is not*

*important for what we want to discuss in this section.*

*Responses to public comments*

**3) Responses to Hui Feng comments**

This is an extended study to the one (Quilfen and Chapron (2021) to present a thorough assess a set of denoised SLA measurements from three altimeters of J3, S3 and AltiKa, to capture short-scale surface signal.  Well recognized due to low SNR by altimeter data, retrieving fine- scale ocean dynamics thus requires preliminary noise filtering.  Smoothing low-pass filters (e.g., running mean, loess or other filters) are often adopted. These filters can smoothen altimeter signals, but maybe result in somewhat loss of small-scale geophysical signals.

Apparently, this EMD approach can be used to design an effective and objective filter to generate denoised surface signals (SSHA, SWH, etc.).   I have a few comments and suggestions on this manuscript.

- There are duplicated contents in the method section very similar to the one (Quilfen and Chapron 2021). Maybe need a rewriting in the revision.

*This ESSD paper aims to publish the denoised SLA dataset, which is the main focus here rather than the presentation of the method. The methodology section is necessary*

*for a proper understanding of the results and to assess the specificity of the approach, and it refers to the paper by Quilfen and Chapron (2021)  which is entirely devoted to the EMD denoising method. Note that the 2021 paper presents the EMD denoising method applied to significant wave height while it is applied to sea level in this manuscript with some improvements (e.g. determination of the A parameter). However,*

*as suggested by Hui Feng, we can indeed shorten this section and we have thus moved a part of the method description in an Appendix..*

- The paper at page 510 states "the adjustment of the EMD denoising process for Jason-3 and Sentinel-3 was performed by using the AltiKa results as reference". Is this objective way to do such an adjustment

*The reason for such an adjustment is explained on page 17. Rather than discuss whether this is an objective way to proceed, we can argue that it is our choice to provide a combined data set of three altimeters showing consistency in their mean power spectral density. AltiKa and Sentinel-3 show very similar noise content and PSD shape, so the approach is easily justified. This is not the case for Jason-3 where the sea level*

*measurements are significantly noisier. However, the strength of the methodology also lies in the fact that each denoised data is attached with its locally estimated uncertainty. Therefore, Jason-3 denoised measurements are attached with larger uncertainty estimates. We believe this is an advantage for ocean modelers, for example, to have both consistency in PSDs and a supply with a meaningful uncertainty estimate.*

• It looks EMD, a data-driven method, leads to a more complicated design process for any specific applications, such as a specific coast/shelf region. The two-step analysis should be completed using regional altimeter data to determine an optimal value of A. In addition, One single value of A in a region may not well represent the seasonality. Look forward to seeing any comments on this by the authors.

*This is not the case, the EMD denoising process does not lead to a different or more complicated design process for a specific application. It can be applied in the same way, with the same algorithms for different data sets (SSH, SWH,Sigma0) and different environmental conditions or ocean basins, including coastal areas. This is because it is a data-driven, self-consistent method that relies on local noise estimation and signal*

*thresholding if the signal to noise ratio is too low, see Equations 3 and 4. In Equation 4, the main term that makes the rule is En, the locally estimated noise energy. A is only a global parameter to allow a global adjustment of the method according to criteria to be defined by the developer. We detail how we derived the value of A, and show that one can indeed obtain slightly different values if one varies the data set used to tune it,*

*but this can be considered as a kind of noise on the estimation of A. So we chose an average value that can be used everywhere, every time.*

• It will be nice if the datasets used in this paper, and processing codes can be available for the readers and potential users who are interested in using the EMD

*The dataset is already available, as indicated in section 5: Data availability, and the*

*Matlab code is now available as indicated in the new Code availability section..*

**4) Responses to Sergei Badulin comments:**

The paper addresses the burning problem of extending results of altimetry to studies of Ocean variability. The authors combine techniques of empirical mode decomposition and adaptive noise filtering to generate noise-free data to better resolve scales 30-100 km. These scales contain essential information on ocean dynamics, first of all, on coupling waves and currents. The new approach provides a useful tool to attack many problems of mesoscale dynamics of

Ocean dynamics. Additionally, the authors are developing an open-access database https://doi.org/10.12770/1126742b-a5da-4fe2-b687-e64d585e138c for the altimetry community to play with.

The paper is essentially based on Quilfen Y., Chapron B.Advances in Space Research, 68. https://doi.org/10.1016/j.asr.2020.01.005, 2021. In my opinion, it would be useful to accompany the text with extractions from the cited paper or to provide as an appendix a scheme of the de-noising technique in a spirit of sect.4.2.

*Thank you for reading and commenting our manuscript. We agree that we can do better to clarify the methodology section and focus on the main results. We have therefore added an Appendix in the revised manuscript.*

**5) Responses to Lotfi Aouf comments:**

First I congratulate the authors for this work. After using of EMD technique to denoise Significant wave height from altimeters, this work describes the implementation of such technique to improve sea level shorter mesoscale up to 30 km of wavelength. The methodology is clearly detailed and the application to 3 altimeters missions is successfully achieved. The spectral analysis clearly indicated the improvement of the power spectrum of sea level for small scale. the implementation of this technique seems easy to set with effort of explanation from the authors, however it is difficult to follow the process of threshold, iterations and ensemble.

When the authors say "ensemble" it means portion of the data ? I need to clarify this point because I am a bot confused with ensemble using external random perturbations. This work open a very interesting perspective of better estimate the geostrophic currents for smaller scale. It will be good if the author say few words, which can suggest how far this technique improve the estimate of geostrophic currents.

Overall the paper is well written and open very promising applications for wave-currents interactions.

*The authors are grateful for the positive feedback and are pleased to respond to the two main points raised:*

1) *We understand that our description of the denoising process may be not that clear, as described lines 246-257. We can try to summarize as follows. For a given data segment typically covering several hundreds km along the satellite track, denoising is performed in several steps, the first being an initial EMD expansion of the signal, SGN, into a series of IMF. The first IMF, IMF1, derived from this EMD expansion is the high-frequency component of the signal along the track, mainly associated with the high-frequency noise (HFN). HFN is estimated in a second step by a wavelet analysis of IMF1. As HFN is only a particular realisation of the noise, to make the full denoising scheme more robust, a third step is to generate a set of N=20 new realisations of the noisy signal, SGN(1:20), by random perturbation of HFN, HFN(i=1:20). Each new noisy signal, SGN(i=1:20)= SGN – HFN + HFN(1:20), is then denoised separately using the thresholding process to provide a set, or an ensemble, of N denoised signals. Finally, the result is the average of the ensemble of denoised signals and the standard deviation provides an estimate of the uncertainty which is function to the actual signal to noise ratio. Note that random noise perturbation, to provide new noise series, is performed in windows of about 140 km width, as the noise statistics are unlikely to be stationary along-track, mainly due to their dependence on the significant wave height. Hope this makes things clearer.*

2) *Improving the regularity of sea surface height estimates will certainly help the determination of along-track gradients. It shall then serve to better estimate the across-track geostrophic balance. It will also provide more reliable statistics, e.g. occurrences*

*of large along-track SSH gradients. The impacts of these large SSH gradients can then be possibly traced, and more reliably related to sea state SWH gradients. At high latitudes, where the altimeter tracks intersect, these more regular SSH estimates can be combined to map 2D field, leading to better resolved geostrophic surface currents. However, the proposed technique will improve the observation of both balanced and un balanced motions.*